

# Mountain wave and downslope winds impact on wind power production

Kine Solbakken[1], Eirik Mikal Samuelsen[1,2], and Yngve Birkelund[1]

[1]UiT, The Arctic University of Norway, Tromsø, Norway
[2]Norwegian Meteorological Institute, Tromsø, Norway

**Correspondence:** Kine Solbakken (kine.solbakken@uit.no)

**Abstract.**

Vertically propagating mountain waves, accompanied by strong downslope winds, occur frequently along the coast of Norway and cause accelerated surface winds on the lee side and downstream of the mountain. Mountain waves form when stably stratified air flows over a mountain and can potentially impact the power production in wind parks located in complex terrains. Although mountain waves and downslope windstorms have received significant attention within the meteorology community, they have received less focus within the wind energy industry. Taking advantage of wind and power production data from a grid of 67 wind turbines spread across two nearby mountains, this study documents accelerated wind speeds and enhanced power production on the lee side of the mountains compared to at the mountain crest. The result of this study suggests that considering mountain waves in the planning phase of future wind parks may allow for an optimal layout of the wind turbines and improve the profitability. The non-dimensional mountain height $\hat{H}$, is a key parameter for describing the development of mountain waves, and this study finds a strong relationship between $\hat{H}$ and the accelerated downslope winds. The results of this study suggest that mountain-wave-induced accelerated downslope winds tend to occur in the wind park when $\hat{H} < 3$, above this value, the airflow is more likely to be blocked and diverted around the barrier. Finally, the Weather Research and Forecasting model reproduces the spatial variations in the wind speeds within the two wind parks relatively well during periods of strong downslope winds and blocking. However, the differences in the wind speeds at the windward side, the mountain top, and the lee side, are not as pronounced as in the observations.

## 1 Introduction

Onshore wind energy, as one of the most cost-effective renewable electricity sources, will play a key role in decarbonisation, and a rapid global growth is anticipated (Abdelilah et al., 2024). With nearly 25% of the continental areas of Earth being in mountainous terrain (Zhao and Li, 2015), increasing the knowledge of how wind power production can be affected by terrain induced wind patterns is crucial to maintain and further reduce the cost of wind energy (Veers et al., 2019). Mountain waves occur in mountainous areas all over the world (Klemp and Lilly, 1975). Along the Norwegian rugged coastline, with mountain tops, ridges, and steep slopes, mountain waves are frequently reported (Sandvik and Harstveit, 2005; Wagner et al., 2017). This research will address mountain-wave-induced strong downslope winds on the lee side of the mountain, and its impact on wind-





power production. A better understanding of how wind power production can be affected by the mountain wave phenomenon is important to ensure an optimal layout of future wind parks, and to improve power production forecasts.

Mountain waves can form when stably stratified air flows over a mountain barrier (Holton and Hakim, 2013). Stably stratified air resists vertical movement, and when air is forced over a mountain, buoyancy forces act to restore the air to its equilibrium position. The wave created might be evanescent, but it can also propagate far downwind or high up into the atmosphere with an

increasing amplitude (Holton and Hakim, 2013). Depending on the characteristics of the mountain waves, wind farms can be affected in several ways. Horizontally propagating mountain waves, also known as lee waves, can affect the power production in wind parks far downstream of the mountain barrier. Stationary or nearly stationary lee waves with a long wavelength can lead to high or low power production at several wind parks simultaneously, while shorter wavelength lee waves can affect the power production at a single wind park, or even just a few turbines within a wind park (Draxl et al., 2021). Fast propagating lee

waves can cause short term imbalances (Draxl et al., 2021). Vertically propagating mountain waves may be accompanied by strong downslope winds (Durran, 1990), creating favourable wind speeds for power production on the lee side of the mountain. Given the right conditions, these accelerated downslope winds can be two to three times stronger than the wind speeds at the mountain top (Jackson et al., 2013). Strong downslope winds can also cause strong wind gusts and turbulence (Klemp and Lilly, 1975) that can alter the wind power production and reduce the lifetime of wind turbines (Kosović et al., 2025). In

addition, the accelerated downslope winds can terminate in a hydraulic jump, further reinforcing the turbulence and potentially lead to locally reversed air flow (Doyle et al., 2000; Gaberšek and Durran, 2004).

Strong downslope winds can form when there exists a critical layer above the mountain that reflects parts of the wave energy back down to the lower parts of the atmosphere (Durran, 1990; Klemp and Lilly, 1975). A critical layer is a layer in the atmosphere where the wind changes in such a way that the cross-barrier flow becomes zero or reversed (Metz and Durran,

2023). A critical layer can also be induced locally when vertically propagating waves with increasing amplitude become unstable and break above the mountain barrier (Durran, 1990). Conditions favouring increasing amplitude and wave breaking, such as a decrease in air density and reduced wind speed with altitude, are well described in the literature (e.g., Sharman et al., 2012; Durran, 2003).

Known for causing extensive wind damage, aviation hazards and exacerbating the spread of wild fires, downslope wind-

storms have received significant attention from the meteorology community over several decades (Smith, 1985; Durran, 1990; Mobbs et al., 2005; Smith and Skyllingstad, 2011; Rögnvaldsson et al., 2011; Metz and Durran, 2023). Within the field of wind energy, downslope winds, and mountain waves in general, have received less attention (Kosović et al., 2025). According to Wilczak et al. (2019), the impact of mountain waves on wind power production, was documented for the first time during The Second Wind Forecast Improvement Project. Subsequent studies based on the same datasets, further studied the mountain

lee wave impact on power production (Draxl et al., 2021; Xia et al., 2021). Draxl et al. (2021) documented large oscillations in the power production due to mountain waves, corresponding to 11% of the rated power of a wind park located downstream of the Cascade Range in the Pacific Northwest of the United States. Sherry and Rival (2015) linked downslope windstorms to potential wind power ramps 50 km downwind of the Rocky Mountains in Alberta, Canada. The Perdigão-field campaign in Portugal addresses wind flow processes in complex terrain relevant for wind power production including mountain waves



(Fernando et al., 2019). Radünz et al. (2021) studied how various static atmospheric stabilities impacted the power production of two wind farms in Brazil situated on a plateau. The study concludes that wind turbines on the leeward side of a plateau tend to have a higher power production compared to the turbines on the windward side under stable atmospheric conditions.

While previous studies have focused on lee waves and wind energy production (Xia et al., 2021; Draxl et al., 2021), this paper will focus on wind power production under the influence of vertically propagating waves accompanied by strong downslope

winds. To the knowledge of the authors, there are no previous studies addressing this issue. This study is based on a dataset of wind observations and power production collected from an array of 67 wind turbines covering two nearby mountains. The dataset allow for documentation of the spatial variation in hub height wind speeds and turbine performance within the wind park during events of mountain-wave-induced downslope winds, as well as periods when the airflow is blocked by the mountain and diverted around it.

Secondly, this study evaluates the ability of the Weather Research and Forecasting (WRF) model (Skamarock et al., 2008) to reproduce mountain waves and downslope winds, as well as the hub height wind speeds and power production during these events. Within the wind industry community, Numerical Weather Prediction (NWP) mesoscale models, in particular the WRF model, are frequently used for research purposes, wind energy assessments and forecasts (Byrkjedal and Berge, 2008; Fernández-González et al., 2018; Davis et al., 2023; García-Santiago et al., 2024). The NWP models offer estimates of the

wind field over large areas, including spatial and temporal variations, and are capable of reproducing hub height wind features in complex terrain (Carvalho et al., 2013; Solbakken et al., 2021; He et al., 2023). WRF and other mesoscale NWP models have also successfully been employed to model various types of mountain waves on different scales, as well as downslope windstorms (Doyle et al., 2000; Sandvik and Harstveit, 2005; Rögnvaldsson et al., 2011; Wagner et al., 2017; Silver et al., 2020; Xia et al., 2021; Draxl et al., 2021; Samuelsen and Kvist, 2024).

In addition, taking advantage of this rather unique dataset, this study will delve into the relationship between the non-dimensional mountain height, $\hat{H}$ and the observed accelerated wind speeds on the lee side of the mountain. Whether or not mountain waves will form and break, depends on factors such as the potential energy required for the flow to pass over the barrier and the kinetic energy available in the air flow. The non-dimensional mountain height represents the ratio of these factors and within the meteorology community, $\hat{H}$ is recognized as a key parameter for describing the development of mountain waves

(Smith, 1989; Jackson et al., 2013). Numerical studies have shown that the non-dimensional parameter, in combination with the aspect ratio of the mountain, is sufficient information to determine the likelihood of a cross-barrier flow to break aloft and create accelerated wind speeds on the lee side (Smith, 1985; Gaberšek and Durran, 2004).

The remainder of this paper is structured as followed: Sect. 2 describes the study area, the observational data, the methods of this study and the configuration of the WRF model. Sect. 3 includes the results and the discussion, as well as two case studies.

A summary of our findings is given in the conclusion in Sect. 4.





## 2 Data and Methods

### 2.1 Study area and observations

Wind observations and power production data have been collected from two wind parks, wind park A and wind park B, located on the large island Kvaløya in Northern Norway. Figure 1 shows the location of the wind parks, as well as the surrounding
terrain. The topography in the surrounding area is characterised by fjords, straights, and islands, as well as mountains, with elevations ranging from sea level (brown) up to 1800 m above sea level (m asl) (light blue). The prevailing wind direction in the wind parks is from the southeast (SE), and the observed yearly mean wind speed, prior to the wind park development, was 7.86 ms$^{-2}$ and 7.39 ms$^{-2}$ at 80 m above ground level (m agl) at A and B, respectively (Solbakken et al., 2021). The SE wind direction is particularly common during the winter months, when the cold inland climate and relatively warm North Atlantic
Ocean current give rise to a pressure gradient in east-west direction and a strong land-breeze (Grønås and Sandvik, 1998). In addition, during winter high pressure systems building up over land, in combination with low pressure systems frequently present over the North Atlantic Ocean (Solbakken et al., 2021), can further reinforce the pressure gradient and the wind field. Mountain waves are expected to occur more frequently during the winter months, when the stability in the lower part of the atmosphere typically is relatively strong at these latitudes (Boy et al., 2019). Stably stratified air, approaching the coast from
the east and southeast, will be affected by the topography and form complex wind patterns such as wakes, gap winds, mountain waves, and strong downslope winds (Samuelsen, 2007). The operator of the wind parks frequently experiences strong winds on the lee side of the mountain during periods of wind from SE, although this has not yet been documented scientifically (Schmid, 2021).

Figure 2 shows a close-up view of the wind parks and the topography, with dark green indicating sea level and yellow
indicating altitudes up to 560 m asl. The black dots and squares indicate the locations of the wind turbines that in total cover an area of approximately 7 km × 7 km. Wind park A (dots) consists of 47 wind turbines, evenly distributed over the mountain at elevations ranging from 300 to 557 m asl. Wind park B (squares) consists of 20 turbines distributed in row-like formations at elevations ranging from 359 to 514 m asl. The wind turbines are of the type Siemens SGRE-DD-130, with a rotor diameter of 130 m, a hub height of 85 m agl, and a rated capacity of 4.2 MW. Observation data with a 10 minute resolution between
4 September 2020 and 24 January 2021 have been used in this study, including wind speed and wind direction taken at hub height, as well as the power production from each turbine.

In order to study how these particular wind parks are affected by mountain waves, only winds from SE are considered, more specifically wind from the sector 120°-165°. This is the prevailing wind direction in the wind park, and the direction where strong downslope winds are expected to frequently occur. The selection of time slots where the wind comes from SE, is based
on the observations at the turbines, and will be referred to as SE-events. For a time period to be considered a SE-event, the wind direction observed at 40 turbines or more, must be within the selected wind sector. In addition, only periods that last for more than 4 hours are considered, and if a period lasts for more than 48 hours, the first 24 hours are considered to be a separate event.





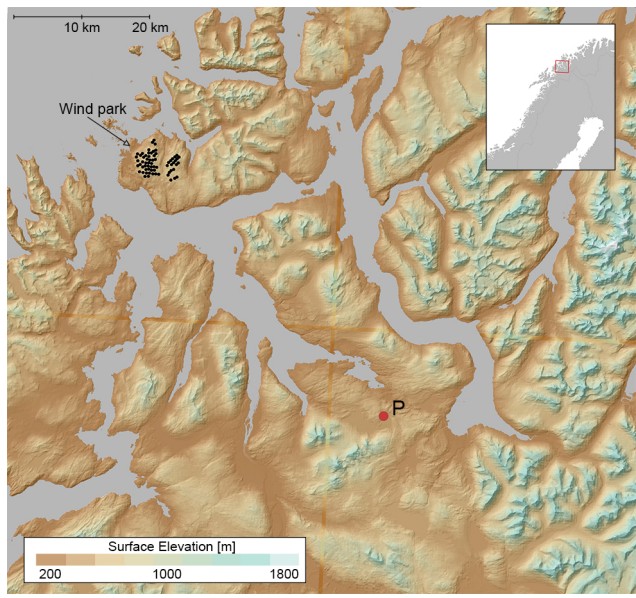

**Figure 1.** The location of the wind turbines (black dots) included in this study, and the surrounding terrain elevation ranging from sea level (brown) to about 1800 m asl (light blue). The map is based on the basic nationwide digital terrain model, with a 10-metre resolution, distributed by the Norwegian Mapping Authority. The red dot indicates the location P where the ERA5 data are retrieved. Inserted map in right hand corner: The land area (grey) and the ocean (white) of the Northern Scandinavian Peninsula. The red rectangle outlines the area of the larger map.

.

The marked areas in Fig. 2, show selected clusters of wind turbines that under SE-wind directions are located upstream of the mountain top (A1 and B1), at the mountain top (A2 and B2), and downstream of the mountain top (A3 and B3). For the purpose of this study, the mean values of the wind speed, wind direction, and power production of each turbine cluster are considered. By evaluating the mean values of the wind turbines within each cluster, instead of values from single turbines, the impact from small local topographic effects is reduced. In the case studies in Sect. 3.3.1 and 3.3.2 wind and power data from each of the 67 turbines of the wind parks are included.

## 2.2 Evaluation of upstream atmospheric conditions

The formation and the development of mountain waves depends on the atmospheric conditions upstream of the mountain. In order to evaluate the state of the atmosphere upstream of the study location, and as an alternative to in situ observations, meteorological parameters have been retrieved from the fifth-generation atmospheric reanalysis (ERA5) provided by the European Centre for Medium-Range Weather Forecasts (Hersbach et al., 2020). The global reanalysis is produced by combining weather observations with numerical weather prediction modeling. With a spatial resolution of approximately 31 km, and hourly out-





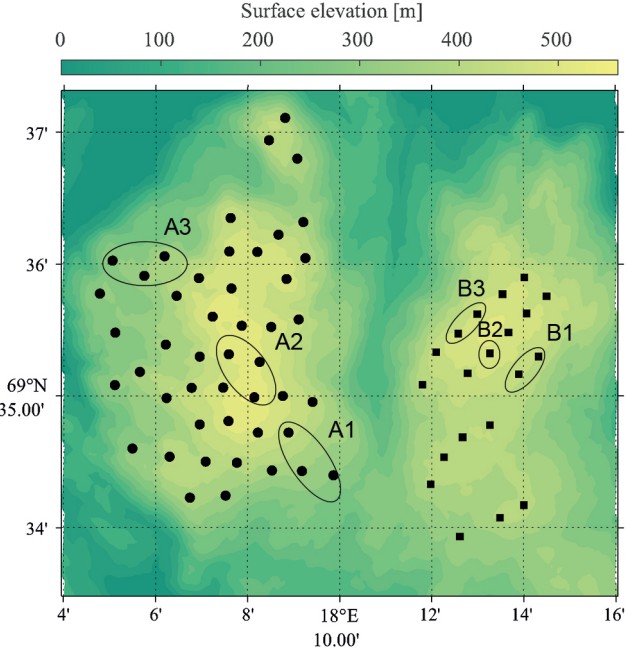

**Figure 2.** The topography and the locations of the wind turbines within wind park A (black dots) and wind park B (black squares). The ellipses mark the turbine clusters located upstream of the mountain tops (A1 and B1), on the mountain tops (A2 and B2), and downstream of the mountain tops (A3 and B3) during SE-events. The map is based on the basic nationwide digital terrain model with a 10 m resolution, distributed by the Norwegian Mapping Authority.

put, ERA5 provides detailed information of the evolution of the atmosphere. Relevant meteorological parameters at the 137 vertical model levels of ERA5 have been retrieved at the model grid point location indicated by a P in Fig. 1.

The Scorer parameter indicates how a mountain wave develops depending on the atmospheric stability and the wind speed, and is given by the following equation:

$$l(z)^2 = \frac{N(z)^2}{U(z)^2} - \frac{1}{U(z)}\frac{d^2U(z)}{dz^2} \tag{1}$$

where the Brunt-Väisälä frequency $N(z)$ and the wind speed $U(z)$ are a function of the altitude $z$. The Scorer parameter is calculated at each ERA5 model level. The horizontal wind speed is decomposed such that only the wind speed in the SE-wind direction of 135° is considered and will be referred to as the tangential wind speed. For simplicity, the curvature term has been omitted in the calculations, so $l(z) = N(z)/U(z)$. When the Scorer parameter $l$ decreases strongly with height, the formation

of trapped lee waves can be expected (Jackson et al., 2013). Favorable conditions for vertically propagating mountain waves are found when the $l$ is nearly constant with height. A Scorer parameter that increases with height indicates conditions allowing for the mountain wave amplitude to grow until breaking occurs. If $l$ increases abruptly with height, as when the cross-barrier flow





becomes zero, or reversed, a critical layer exists where vertically propagating mountain waves can be absorbed and reflected, causing strong downslope winds.

Another parameter frequently used to describe the development of mountain waves is the non-dimensional mountain height denoted $\hat{H}$. This parameter combines the cross-barrier wind speed, the mountain height, and the stability in terms of the Brunt-Väisälä frequency as follows:

$$\hat{H} = \frac{h_0 N}{U} \tag{2}$$

where $h_0$ is the mountain height and based on the elevation of the real terrain within the A2-turbine cluster (Table 1) is set to
be 550 m. It is assumed that the airflow that interacts directly with the mountain is spanning from the surface and up to a height of about 1 km. The Brunt-Väisälä frequency, $N$, is therefore calculated between the lowest ERA5 model level (about 310 m asl) and model level number 17 (varying between 930-1000 m asl). The tangential wind speed $U$ is calculated from the wind speeds at the tenth model level. With heights varying between 553-586 m asl, the tenth model level is the level closest to the real height of the mountain of wind park A. If the non-dimensional height $\hat{H} << 1$, the flow will pass easily over the mountain,
however there will not be any wave breaking causing strong downslope winds. Opposite, if $\hat{H} >> 1$, a stagnation point will occur on the windward side, and the flow will be deflected around the mountain. When $\hat{H} \sim 1$, and the flow is normal to the mountain ridge, a stagnation point is formed above the mountain, where the horizontal wind is much lower in comparison to the wind at lower levels, causing the vertically propagating mountain waves to break (Smith, 1989).

### 2.3 Model and simulation design

For the purpose of this study, the WRF model version 4.3 (Skamarock et al., 2008) is configured to include three two-way nested domains, D01, D02, and D03, with horizontal resolutions of 10.5 km, 3.5 km, and 750 m, respectively. The locations of the domains can be seen in Fig. 3. D01 and D02 consist of $101 \times 101$ and $112 \times 112$ grid points, respectively. D03 consist of 121 grid points in north-south direction, and 146 grid points in east-west direction. Static fields, with 30 arc-second resolutions are applied. The static fields are retrieved from the NCAR database and provided by the 20-category Moderate Resolution
Imaging Spectroradiometer (MODIS) and the Global Multi-resolution Terrain Elevation Data 2010 (GMTED 2010). Table 1 summarizes the real range of elevations within each turbine cluster of wind park A, along with the corresponding model elevations in D03. The vertical structure of the model consists of 51 terrain following sigma levels, with the lowest five levels located below 100 m agl, and the upper boundary at 50 hPa.

The physical configuration of the model consists of the Thompson microphysics scheme (Thompson et al., 2008), the
Rapid Radiative Transfer Model for Global applications (RRTMG) scheme for long and short wave radiation (Iacono et al., 2008), the MYJ surface layer scheme (Janjić, 1994), the Unified Noah Land Surface Model (Chen et al., 1997), the Mellor-Yamada Nakanishi Niino (MYNN) planetary boundary layer scheme (Nakanishi and Niino, 2009), and the Tiedke cumulus parametrization scheme (Tiedtke, 1989; Zhang et al., 2011). For D03, the cumulus scheme was turned off, due to the high resolution, to allow the model to resolve the convective processes explicitly. Wind turbine parametrisation is applied by the

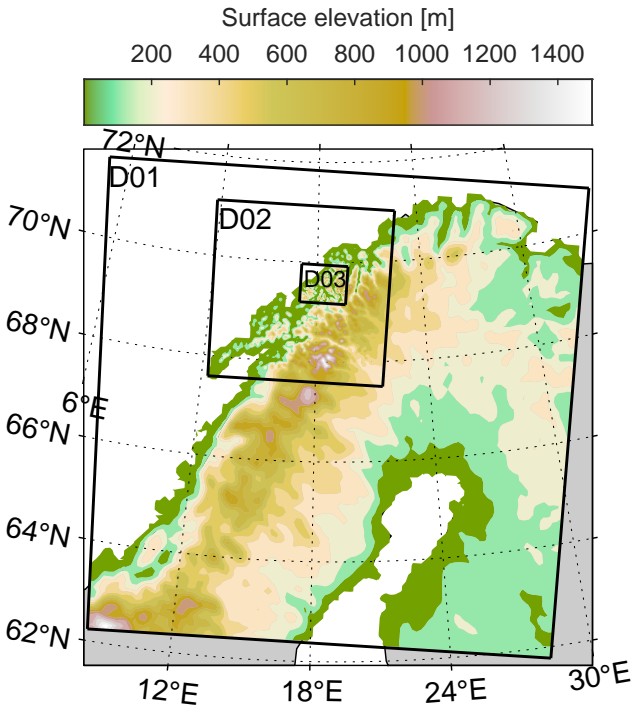

**Figure 3.** The WRF domain configuration, D01, D02 and D03, with ocean in white and the terrain elevations within each domain, ranging from sea level (green) to 1500 m asl (white).

Fitch scheme (Fitch et al., 2012). The power curve and thrust coefficient included are specific for the wind turbines in the park, with the default correction factor of 0.25, as suggested by Archer et al. (2020).

   The ERA5 global reanalysis is used as initial and boundary conditions for the simulations. The simulations are run for 8 days, with the first 12 hours considered spin-up time. The next 12 hours of the simulation period has been interpolated with the last 12 hours in the previous simulation, to allow for smooth overlap of the time series. The simulations cover the period from
4 September 2020 to 24 January 2021. The wind data are retrieved from 85 m agl by vertical interpolation of the model levels, and from the given turbine locations by horizontal bilinear interpolation between the grid points. Similar to the observations, the cluster wind values, and power production are the mean values of each parameter at the turbines within each cluster.

## 3    Results and discussion

For simplicity, this study only evaluates wind coming from the SE-sector 120°-165°. The observations from the A2 cluster
indicate that the wind direction is within this SE-sector for about 35-39% of the study period. Furthermore, time slots referred to as SE-events have been selected as described in Sect. 2.1. Within the study period 67 SE-events have been selected, and combined these consist of 1104 hours of data, corresponding to 32% of the total study period. In the remainder of this study all results presented and discussed are only related to the SE-events.



**Table 1.** The altitude variations at each turbine cluster based on 10 m resolution terrain data and model elevation data collected from domain D03. The values given in m indicate the lowest and highest altitudes of the turbine locations within each cluster.

|    | Real altitude [m] | Model altitude [m] |
|----|-------------------|--------------------|
| A1 | [324-455]         | [304-389]          |
| A2 | [522-551]         | [501-521]          |
| A3 | [308-391]         | [276-409]          |

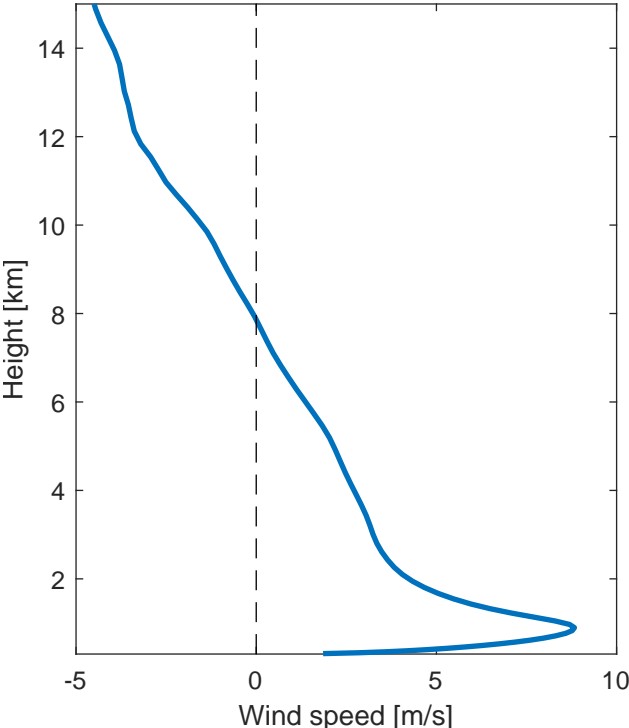

**Figure 4.** The vertical profile of the tangential ERA5 wind speed at location P averaged over all SE-events with wind speeds along x-axis and the average altitude above sea level along the right y-axis.

### 3.1 Non-dimensional mountain height

This section investigates the relationship between the upstream weather conditions at location P (Fig. 1), calculated from ERA5 data, and the accelerated downslope winds in wind park A. The Scorer parameter and how this parameter changes with height, indicate which type of mountain wave will develop during a mountain wave event. A cross-mountain wind decreasing with height, i.e. a reversed wind shear, is a condition where the Scorer parameter increases with height (Eq. 1). In addition, if the cross-mountain flow becomes zero and further reversed, the Scorer parameter goes to infinity with increasing height. Figure



4 illustrates the vertical profile of the tangential ERA5 wind speed at point P averaged over all SE-events, with the tangential wind speed being the component of the horizontal wind in the SE-direction. It is apparent that there is on average for all 67 SE-events a reversed wind shear from about 900 m asl in the cross-mountain flow, and a mean state critical layer at just below 8000 m asl. This distinct signature in the upstream large scale wind profile is quite striking and indicates upstream weather conditions in which both mountain waves might grow and break, as well as being dissipated at a critical level.

Figure 5a) illustrates the relationship between the downslope winds at A3, and the non-dimensional mountain height $\hat{H}$. The y-axis shows the A3 wind speed values normalised with respect to the wind speed at A2. The cluster wind speeds are the mean value of the observed nacelle wind speeds at the three turbines within each cluster, as defined in Sect. 2.1. A normalised wind speed above one, means the wind speed is higher at the lee side than at the mountain top. The x-axis shows the values of $\hat{H}$ calculated from ERA5 data extracted at the upstream location P. The green dots show the hourly values, while the thicker blue dots represent the same values averaged over the period of each SE-event. During the SE-events studied, the wind speeds at the lee side of the mountain are typically stronger than the wind speed at the mountain top. When considering the hourly values of all SE-events, the wind speed at A3 is higher than at A2, 80% of the time. In addition, 18% of the time, the wind speeds are 1.5 times higher or more at A3 compared to A2. The non-dimensional mountain height varies mainly between 0 and 5 during the SE-events considered. In addition, there are 10 sporadic timestamps (green dots), with $\hat{H}$ values above 5, that are not included in the figure. As $\hat{H}$ increases from zero to a value of about 1.5, there is a tendency of an increasing normalised wind speed. When $\hat{H}$ increases further, the normalised wind speeds decrease. As $\hat{H}$ increases above 3, the normalised wind speeds are typically below one. These results correspond well with previous studies such as Smith (1989) and Gaberšek and Durran (2004). Gaberšek and Durran (2004) studied idealised airflow over a mountain barrier and found that for $\hat{H}$ lower than 0.25, mountain waves are created over the barrier, but no wave breaking is present, resulting in only slight enhancement of the wind speed downstream of the barrier. When the wind speed is increased such that $\hat{H}$ equals 1.4, the wave breaks over the ridge and the wind speed increases in a narrow zone downstream of the barrier. In Gaberšek and Durran (2004) there is partial blocking for $\hat{H} = 2.8$, and full blocking for $\hat{H} = 5.0$. The current study shows a similar pattern with blocking at A3 occurring for $\hat{H}$ above 3 according to Fig. 5.

Figure 5a) displays a strong relationship between $\hat{H}$ and the A3 wind speeds, and for this particular wind park, the non-dimensional mountain height is to some extent able to indicate when to expect enhanced lee side winds due to mountain waves, and when to expect lower wind speeds on the lee side of the mountain due to blocking. The single parameter calculated from readily available meteorological data can hence be useful both in planning and for power prediction purposes. The strong relationship between $\hat{H}$ and the A3 wind speeds suggests that breaking of internal gravity waves are the dominant mechanism responsible for the strong downslope winds in wind park A. As described in Sect. 2.2, $N$ in Eq. 2 is estimated from ERA5 model levels spanning from the surface and up to a height of about 1 km, and $U$ from the ERA5 level closest to the mountain peak. The sensitivity of the results to the selection of vertical levels is tested by shifting two levels upward and two levels downwards. In addition, small variation in the value of $h_0$ in Eq. 2 is also tested. The results appear to be only weakly sensitive to the choice of vertical levels and the value of $h_0$, with the overall findings remaining consistent. One of the reasons why the rather simple theory of the non-dimensional mountain height appears to hold in this real-world case, as opposed to an idealised





model, may be the long and well-defined fjord that channels the wind from location P towards the mountain. Another reason may be that the lower-resolution ERA5 data, as opposed to local observations or high-resolution WRF simulations, provide a mean state of the atmosphere free of local terrain effects at location P.

Although the result in general aligns with the theory of the non-dimensional mountain height, the scatter plot also reveals some deviating results. The relationship between the non-dimensional mountain height and the accelerated downslope winds is therefore further evaluated by normalising the A3 wind speeds with respect to the A1 wind speeds. The resulting scatter plot in Fig. 5b) shows similar tendencies as the scatter plot in Fig. 5a). However, for nine events (indicated by blue-circled markers) with $\hat{H} > 1$, the mean normalised wind speeds change from values below unity in Fig. 5a) to values exceeding unity in Fig. 5b). The results suggest that strong downslope winds are present also in these nine events. However, the isentropic surfaces may be compressed also above the mountain crest, rather than solely on the lee side, resulting in an area of accelerated winds that includes both A2 and A3.

Regardless of the method used to decide the relative wind speed at A3, there are still two events (blue dots enclosed in squares) that do not align with the theory, and further analysis is conducted based on the WRF simulations. In both events, the simulated wind field at 85 m agl (not shown) are characterised by large spatial and temporal variations within the DO3-domain. In general, the wind speeds are low over the mountains, while higher wind speeds are present in the fjords and the valleys. The wind directions within the domain show large spatial variations with winds from the southeast as well as from the northeast. In addition, the simulations show periods where the winds through the wind park area are influenced by mountain waves originating from the terrain far upstream. At location P, the WRF simulations reveal wind directions shifting throughout the time periods of the two events, and on several occasions have a northerly component. This suggests that under certain weather conditions, location P may not be representing the upstream conditions of the wind park. In future studies, the relationship between $\hat{H}$ and the A3-wind speeds could be further investigated by including upstream meteorological data from multiple sources and locations.

## 3.2 Downslope winds and impact on power production

The impact of mountain waves and strong downslope winds on the wind power production is evaluated in terms of the wind-speed distribution and the power production at the three wind turbine clusters. Figure 6a) shows the variations in the observed wind speed distribution between the clusters A1 (blue), A2 (green), and A3 (red) when only the SE-events are included. The histogram bins have been divided into intervals of 0.5 ms$^{-2}$ along the x-axis and the frequency on the left y-axis are given in 10-minute values converted to hours. In order to allow for a comparison of the wind speeds and the potential power production, the power curve for the turbines have been included in the same figure (black dashed line) with power along the right y-axis. The power curve consist of four Power Curve Zones (PCZ): PCZ1 with wind speeds below the cut-in threshold and no power production, PCZ2 with wind speeds between the cut-in wind speed and the rated wind speed, i.e. where the power increases from 0 and up to the rated power, PCZ3 where the turbines produce at the rated power until cut-off wind speed, and PCZ4, the zone where the turbines are shut down due to high wind speeds. For the turbines in these particular wind parks, the cut-in wind speed is at 3.0 ms$^{-2}$, the cut-off wind speed is at 28.5 ms$^{-2}$, and the rated wind speed is 14.5 ms$^{-2}$.





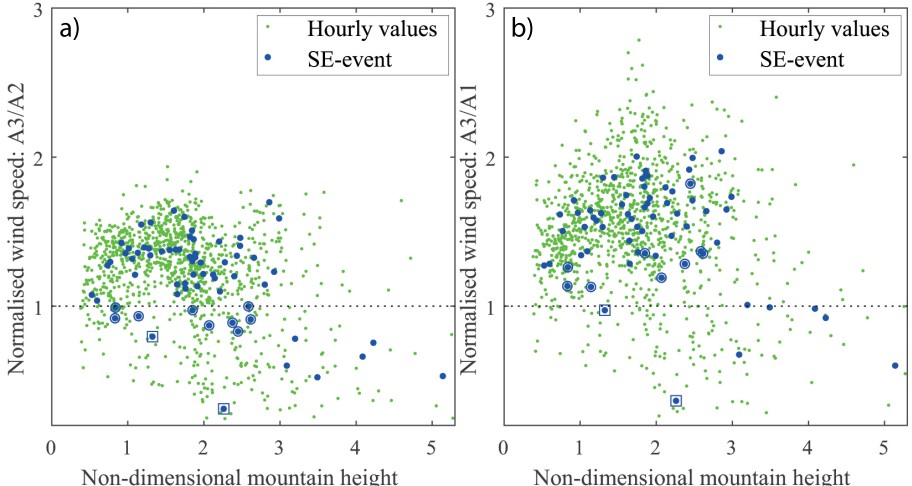

**Figure 5.** Scatter plots showing the hourly values (green dots) of the non-dimensional mountain height and the normalised wind speeds at A3. The blue dots indicate the average values taken over each SE-events. Blue dots enclosed in circles and squares are similar to the blue dots and further details are given in the text. a) A3 wind speeds normalised with respect to A2 wind speeds. b) A3 wind speeds normalised with respect to A1 wind speeds.

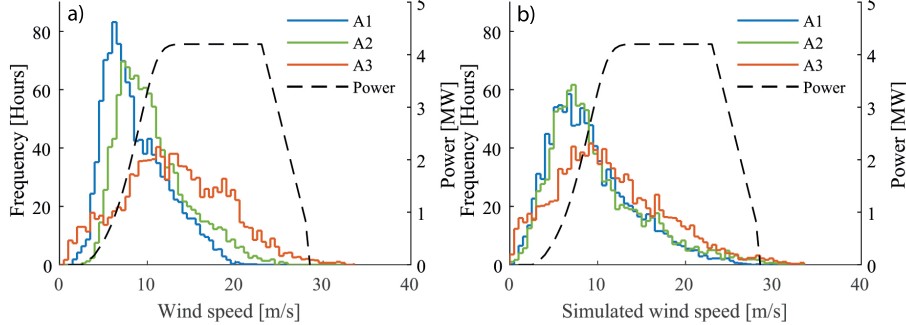

**Figure 6.** a) Observed wind speed frequency distributions for the turbine clusters at A1 (blue), A2 (green), and A3 (red), with wind speeds along the x-axis and wind speed frequency in hours along the left y-axis. The dashed line represents the turbine power curve, and the values are indicated along the right y-axis. b) Similar to a) but representing the simulated wind speeds.

The A3 wind cluster has a higher frequency of wind speeds favourable for wind power production in comparison to A1 and
270 A2. While the A1 and A2 wind speed distributions have similar shapes, shifted towards the left, with peaks at approximately
6 ms$^{-2}$ and 7 ms$^{-2}$, respectively, the A3 wind speeds are more evenly distributed with a peak around 11 ms$^{-2}$. At A1 and A2,
the frequency of wind speeds within PCZ2 is considerable higher than at A3. Consequently, the A1 and A2 turbines are more
often producing below the rated power compared to A3. In addition, small variations in wind speed within PCZ2, results in
large variations in the power production. At A3, on the other hand, the frequency of wind speeds within PCZ3 is considerably
higher than at A1 and A2, hence the A3 turbines will more often produce at their maximum power. In addition to the enhanced





**Table 2.** The observed and the simulated capacity factor $C_f$ and mean wind speed $\overline{U}$, as well as Bias, MAE, and the correlation coefficient R.

|    | $C_{f,Obs}$ | $C_{f,WRF}$ | $\overline{U}_{Obs}$ | $\overline{U}_{WRF}$ | Bias | MAE | R |
|----|------|------|-------|-------|-------|------|------|
| A1 | 0.48 | 0.52 | 8.68  | 9.25  | 0.57  | 2.73 | 0.68 |
| A2 | 0.60 | 0.55 | 10.57 | 9.85  | -0.72 | 3.19 | 0.67 |
| A3 | 0.73 | 0.64 | 13.15 | 11.38 | -1.77 | 4.50 | 0.61 |
| B1 | 0.54 | 0.56 | 9.51  | 9.79  | 0.28  | 2.95 | 0.66 |
| B2 | 0.66 | 0.61 | 10.75 | 10.26 | -0.49 | 3.23 | 0.65 |
| B3 | 0.71 | 0.61 | 12.26 | 10.48 | -1.74 | 3.68 | 0.62 |

power production on the lee side of the mountain of wind park A, the same wind turbines record wind speeds above the cut-off threshold, and below the cut-in threshold, more often than the turbine clusters upstream. While the high wind speeds can be linked to mountain wave-induced downslope winds, the occurrence of low wind speeds at A3 can be due to blocking of the air flow. The large difference in the wind speed distribution between A3 and the two other clusters, has a clear impact on the
wind power production and is expressed through the capacity factor $C_f$ in Table 2. The capacity factor is a commonly used parameter to evaluate how well-sited a wind turbine is, and for the purpose of this study, $C_f$ is calculated from the power produced during the SE-events only. The A3 turbines are producing considerably closer to its maximum in comparison to the turbines in the two other clusters. During the selected SE-events, the A3 turbines produce 51% and 21% more than the A1 turbine and A2 turbines, respectively.

Similarly, the wind speed distributions (not shown) for the three clusters in wind park B, as indicated in Fig. 2, reveal a higher frequency of higher wind speeds on the lee side, compared to the upstream turbines, although the difference between the clusters is not as pronounced as in wind park A. The capacity factor in Table 2 reflects the spatial variations within wind park B, with $C_f$ being highest at B3 and lowest at B1.

The result of this study emphasises the importance of an extensive understanding of the wind flow in complex terrain during
various atmospheric stability conditions both in the planning phase and for wind power productions forecasts. Although there are no observations of severe wind speeds in these particular wind parks during the study period, downslope windstorms can result in severe and occasionally damaging wind speeds, large wind-speed fluctuations, as well as turbulence and wind gusts (Klemp and Lilly, 1975; Durran, 1990), and should be considered in the planning phase.

### 3.3 WRF

The ability of the WRF model to reproduce the observed wind patterns in the wind parks is evaluated in terms of the mean features during the SE-events, such as the wind speed distribution, the mean wind speed, as well as the capacity factor calculated over the same periods. In addition, the ability of the model to reproduce the temporal pattern of the observations is evaluated in terms of the Mean Absolute Error (MAE) and the correlation coefficient (R). These statistical parameters and the Bias are also





provided in Table 2 and are calculated as described in Solbakken et al. (2021). In addition, the ability of the WRF model to
reproduce the wind directions during the SE-events is evaluated in terms of wind roses (not shown) for all the 9 turbines within
the clusters in wind park A. WRF is able to reproduce the wind directions with only small deviations from the observations.

In agreement with the observations, the simulations show a higher mean wind speed at the lee side of the mountain (A3),
compared to the mean wind speeds at the mountain top (A2), and at the upstream location (A1). When evaluating each turbine
cluster separately, the wind speed Bias indicate that the model tends to overestimate the wind speeds at A1, while underesti-
mating the wind speeds at A2 and A3. The negative Bias at A3 is considerably larger than at A2. The WRF model is also able
to reproduce the spatial variations in power production, with the highest $C_f$ at A3 and lowest $C_f$ at A1. Although the model is,
to some extent, able to reproduce the relative power production of the three turbine clusters, the increase in $C_f$ from A1, to A2
and A3, is not as pronounced as what is seen in the real $C_f$. Similar wind speed biases are found in the clusters of wind park
B. Although the model is able to reproduce the higher mean wind speed at B3, compared to B2, WRF is not able to reproduce
the increase in $C_f$ from the B2 cluster to the B3 cluster.

When comparing the temporal patterns of the simulations and observations, the MAE is lowest at A1, higher at A2, and
highest at A3. The correlation coefficient is the highest at A1, and the lowest at the A3 cluster. Wind park B exhibits similar
patterns in MAE and R (Table 2), with higher error and lower correlation at the B3, compared to the turbine clusters upstream.
The reduced accuracy in the temporal pattern at the lee side turbines, is also seen in the study by Rögnvaldsson et al. (2011), and
can be linked to the increased complexity of the wind patterns from the windward side to the lee side due to mountain waves.
The MAE and the R of the current study are of similar values as in previous studies conducted in the same area (Solbakken
et al., 2021; Solbakken and Birkelund, 2018). In addition, the MAE and the R in the current study, with a 750-metre horizontal
grid resolution, are slightly improved compared to what was found in the 1 km simulations by Solbakken et al. (2021) at
the same location. However, the results are not directly comparable due to differences in the model configurations and study
periods.

Figure 6b) presents the wind speed distributions of the simulated wind speeds at the three turbine clusters in wind park A.
The WRF model is able to reproduce the observed differences between the A3 wind speed distribution, and the distributions
of the two other clusters. In agreement with the observations, there is a high occurrence of wind speeds above 12 ms$^{-2}$ at A3,
including wind speeds above the cut-off threshold, while at A1 and A2 there is a lower occurrence of the same wind speeds.
The higher occurrence of the PCZ3 and PCZ4 wind speeds at A3, compared to A1 and A2, indicate that the model is able to
simulate the formation of mountain waves above the wind park causing the accelerated winds on the lee side of the mountain.
For the lower wind speeds, below the cut-in threshold, the model succeeds in reproducing the higher occurrence of these wind
speeds at A3 compared to A1 and A2. This result suggests that the model is also able to reproduce periods where the airflow is
blocked by the mountain and diverted around.

Although the model is able to reproduce the distinct wind speed distributions observed at the three clusters, the distributions
also reveal some shortcomings in the ability of the model to accurately reproduce the complex wind patterns in the wind park.
For instance, the differences between the simulated wind speed distributions of the three clusters are not as pronounced as
they are in the observations. In particular, the A1 and A2 wind speed distributions appear to be more similar in comparison to





the observed wind speed distributions at the same locations. When comparing the simulations with the observations at the A3
cluster, it is apparent that the model underestimates the high occurrence of the PCZ3-wind speeds. The underestimations of
these higher wind speeds suggests that the model is not fully able to reproduce the frequency and the strength of the downslope
windstorms. It is worth noting that for the highest wind speeds at A3, above the cut-off threshold, the simulated wind speeds
are more accurate. At A2 the simulated wind speed distribution skews left in comparison to the one of the observations. In
particular, the model overestimates the frequency of the lower wind speeds and underestimates the higher wind speeds. At A1,
the model considerably underestimates the lower PCZ2 wind speeds, while overestimating the occurrence of the wind speeds
within the PCZ3. The shift in the simulated wind speed distribution, with under- and overestimation in the PCZ2 and PCZ3,
respectively, agrees well with previous studies conducted in the same area (Solbakken and Birkelund, 2018; Solbakken et al.,
2021). However, for the lower wind speeds, below the cut-in threshold, the model overestimates the frequency at all three
turbine clusters. A similar overestimation of the lower wind speeds are not seen in the studies of Solbakken and Birkelund
(2018); Solbakken et al. (2021).

Several factors may impact the accuracy of the model simulations. For the simulations of mountain waves and downslope
windstorms, an accurate representation of the upstream flow properties is particularly important. Rögnvaldsson et al. (2011)
highlights the importance of the micro-physical processes in the formation of downslope windstorms and found that the Thompson micro-physics parameterization scheme outperformed the five other schemes tested in the study. Notably, Rögnvaldsson
et al. (2011) found that the Thompson scheme simulations showed a stronger upstream stability at, and above, the mountain
barrier, compared to the simulations run with the other schemes. Furthermore, Rögnvaldsson et al. (2011) found that simulations run without parameterization of the micro-physics processes and surface fluxes, resulted in stronger wind speeds on the
lee side of the mountain barrier. A higher horizontal resolution, hence, a better representation of mountain peaks in particular,
is also expected to improve the accuracy of simulated meteorological metrics during mountain wave events (Samuelsen and
Kvist, 2024; Wagner et al., 2017). In the D03 domain of the current study, the overall terrain is replicated well, with the model
terrain only slightly lower than the real terrain (Table 1). Although, the mountain peaks appear to be sufficiently represented in
the model, a higher resolution could improve the representation of the mountain shape, which also plays a crucial role in the
development of mountain waves (Durran, 1990). In addition, some smaller scale terrain features are missing in the model and
may impact the accuracy of the simulations.

Another factor impacting the wind flow within the park, is the wakes created by the wind turbines. Studies suggest that under
stable conditions, the strength of the wakes is higher and the recovery slower, than during unstable and neutral conditions (Han
et al., 2018). For the purpose of the current study, the Fitch-wind farm parameterization scheme is employed in the model to
allow for the impact from the wind turbines on the simulated wind flow. However, the accuracy of wind power parameterization
schemes under stable atmospheric conditions is not thoroughly explored. For instance, García-Santiago et al. (2024) found that
when comparing the mesoscale simulations with large eddy simulations, the Fitch-scheme exhibit larger deviations under stable
atmospheric conditions, than during neutral or unstable atmospheric conditions. In addition, the Fitch scheme does not take
into account mechanical and electrical losses, resulting in an overestimation of the turbulent kinetic energy (TKE) generated by
the turbines in the model Fitch et al. (2012). In order to reduce the TKE source, Archer et al. (2020) suggests the introduction



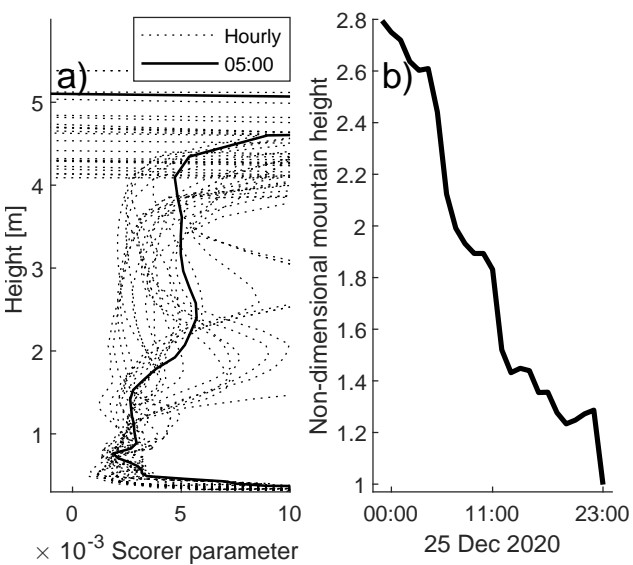

**Figure 7.** Case study 1 on 25 December 2020: a) Hourly Scorer parameter (along x-axis) at altitudes indicated along the y-axis. b) The non-dimensional mountain height $\hat{H}$.

of a correction factor. Due to the lack of a reliable estimate for the correction factor during stable atmospheric conditions, the
initially suggested correction factor of 0.25 (Archer et al., 2020) is employed in the simulations of the current study. However, the study by García-Santiago et al. (2024), suggest that under stable conditions, as compared to neutral conditions, an increased correction factor could improve the simulation results.

This study demonstrates the ability of the WRF model to reproduce spatial patterns of hub height wind speeds relevant for wind power production in complex terrain. It is possible that tuning the model configuration to more optimal settings could
further improve the simulation results, particularly if the stability is affected (Rögnvaldsson et al., 2011). A future study should therefore include a sensitivity analysis, as well as investigate the impact of the wind farm parametrisation on the wind field within the parks during periods with strong downslope winds.

### 3.3.1 Case Study 1: Mountain-wave-induced accelerated wind speeds

The first case study is the SE-event running from 24 December 2020 at 23:20 local time (LT) to 25 December 2020 at 23:20
LT, where a strong downslope wind is observed. Figure 7a) shows the Scorer parameter at every hour (dotted line) during the period considered. At 05:00 LT, the Scorer parameter (solid line) is nearly constant between 1000 and 1500 m asl, and increasing between 1500 and 2500 m asl, indicating conditions where vertically propagating mountain waves may form and break. A critical layer is present at heights from about 4500 m asl. Figure 7b) shows that the non-dimensional mountain height decreases from 2.8 to 1 during the period of the case study. Based on the results in Sect. 3.1, these are $\hat{H}$-values where strong
downslope winds are expected at the A3 cluster.



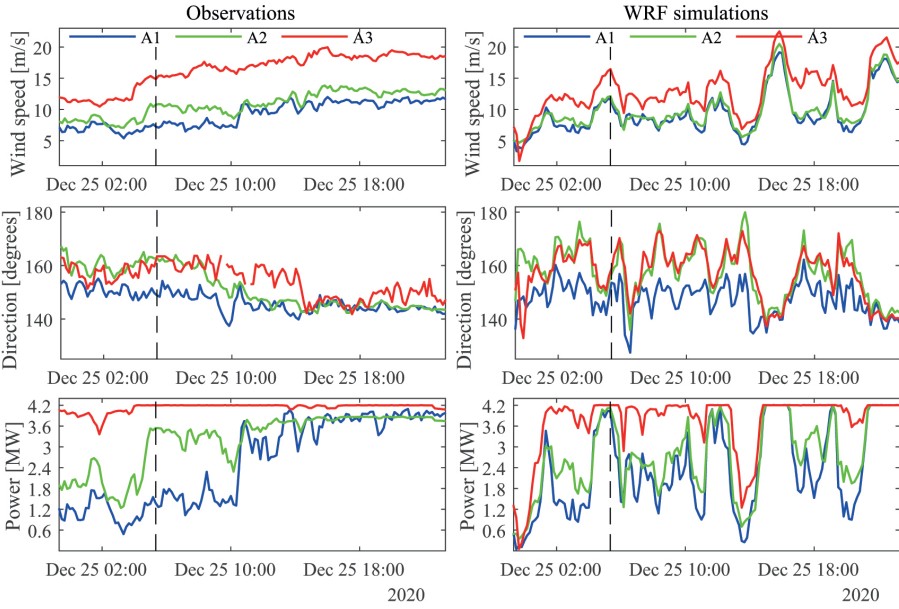

**Figure 8.** Observed (left) and simulated (right) wind speeds (top), wind directions (middle) and power production (bottom) for the three wind turbine clusters A1 (blue), A2 (green), and A3 (red) on 25 December 2020. The vertical dashed lines indicate the time of Fig. 9.

Figure 8 presents the observed (left) and simulated (right) wind speed (top), wind directions (middle) and power production (bottom) during this event for the turbine clusters A1 (blue), A2 (green), and A3 (red). The observed wind directions at the three turbine clusters vary between 140-165°. The observed wind speeds at A3 vary between 11 and 19 ms$^{-2}$, and are persistently stronger than the wind speeds at A1 and A2, where the wind speeds vary between 5 and 13 ms$^{-2}$. Consequently, the A3 turbines

operate at rated power nearly throughout the entire 24-hour period, while the A1 and A2 clusters encounter a considerably lower power production, particularly during the first 12 hours. The simulated wind directions exhibit larger variations within a slightly larger sector, compared to the observations. In agreement with the observations, the simulated wind speed is higher at A3, than at A1 and A2. However, compared to the rather steady observed wind speeds, the simulated wind speeds exhibit large temporal variations. The simulated power production is higher at A3 compared to A2 and A1, although with large oscillations

at all clusters due to the erroneous variations in the simulated wind speeds.

Figure 9 presents the observed and simulated winds on 25 December 2020 at 05:20 LT, indicated with the dashed line in Fig. 8. Figure 9a) shows the observed and Fig. 9b) the simulated wind speeds at hub height (85 m agl) at each wind turbine. The observed wind speeds are ranging from 5 ms$^{-2}$ (blue) to 15 ms$^{-2}$ (orange). The turbines upstream of the mountain top exhibit wind speeds ranging from about 5 to 10 ms$^{-2}$, while the turbines on the mountain top encounter slightly higher wind

speeds. The highest wind speeds of 15 ms$^{-2}$ are found on the lee side of the mountain, particularly within turbine cluster A3. In general, the WRF simulations reproduces the spatial wind pattern apparent in the observations well. However, the simulated

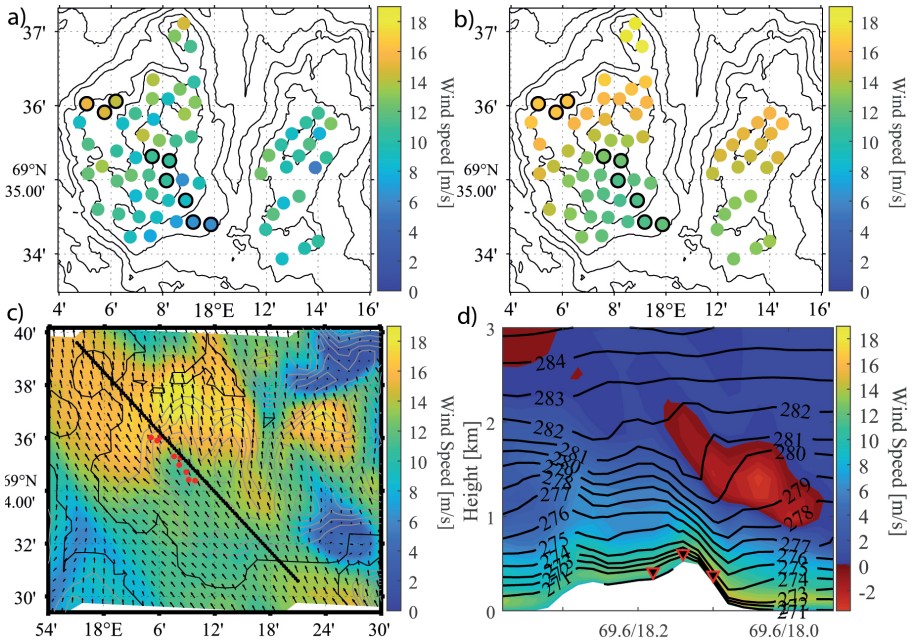

**Figure 9.** a) Observed wind speeds and b) simulated wind speeds at 85 m agl at the turbine locations. c) Horizontal wind speed at 85 m agl. The solid contours indicate the land areas, and the dotted contours indicate the model terrain elevation with 100 m intervals. The red dots indicate the locations of the turbines within cluster A1, A2 and A3. d) Vertical cross section in SE-NW direction, as indicated by the dotted line in the figure c), showing the potential temperature (black lines) and tangential wind speeds. The triangles indicate the location of the turbine clusters in wind park A. All figures are from 25 December 2020 at 05:20 LT.

wind speeds in the wind parks are in general overestimated and have less spatial variations in wind speeds within the wind parks compared to the observations.

Figure 9c) displays the spatial variations of the simulated horizontal wind speeds at 85 m agl over a slightly larger area
than Fig. 9a) and b). The D03 model terrain elevation is indicated by grey dashed contour lines with 100 m intervals, and the solid contour line outlines the land areas. The turbines within cluster A1, A2, and A3, are indicated by red dots. The wind speeds range from 0 ms$^{-2}$ (dark blue) to 19 ms$^{-2}$ (yellow) and the arrows indicate the wind direction. The airflow approaches the mountain from the southeast, with relatively unidirectional flow across the mountain. The wind speeds on the windward side and over the mountain top are of similar magnitude, while on the mountain lee side there is a large area of higher wind
speeds, including the location of the A3 wind turbines.

The vertical cross section of the simulated tangential wind speeds and the potential temperature in Fig. 9d), suggests mountain-wave-induced accelerated winds on the lee side of the mountain. The position of the vertical cross section is indicated by the southeast-northwest oriented dotted line in Fig. 9c) and the tangential wind speed is the horizontal wind speed decomposed in the approximate direction of the cross section. The solid horizontal contour lines represent constant potential
temperature (isentropes) with line spacing of 1 K. The colours blue (0 ms$^{-2}$) to yellow (19 ms$^{-2}$) indicate positive tangential





wind speeds in the SE-direction, while the red colours indicate reversed wind speeds, i.e. a northwesterly component, ranging from -3 to 0 ms$^{-2}$. The terrain is indicated in white, and the three turbine clusters are marked with triangles at their approximate location. The potential temperature is increasing with altitude, and a reversed wind speed shear is present from the surface level and up to about 3000 m asl, in which the wind speed becomes negative, indicating the presence of a critical layer. The

mountain waves are apparent from the isentropes. The growing wave amplitude, the upstream-tilting isentropes, and the locally reversed winds, suggest wave breaking with a self-induced critical layer at about 1000-2000 m asl above the mountain. This is in accordance with the Scorer parameter in point P (Fig. 7). Beneath the stagnant area, the isentropes are compressed, and the wind speeds accelerate towards the surface on the lee side of the mountain with wind speeds exceeding 15 ms$^{-2}$ (orange).

### 3.3.2    Case study 2: Partial blocking

The second case study is on the 11 December 2020 at 08:20 to 19:20 LT, with weaker wind speeds on the lee side than upstream, suggesting blocking of the airflow. In this case study, Figure 10a) depicts a Scorer parameter with large variations over time. At 12:00 LT (solid line), the Scorer parameter is near constant at the lowest levels, and increasing from about 1500 m asl to 2500 m asl, indicating upstream weather conditions favourable for the development of vertical propagating and breaking mountain waves, as well as the presence of a critical layer. However, the Scorer parameter is only relevant when the air flows over the

mountain. In this case study, the non-dimensional mountain height, in Fig. 10b), varies between values above 3, where blocking is expected to occur at A3, and below 3, indicating accelerated downslope winds at A3 according to Fig. 3.1. At 12:00 LT, $\hat{H}$ is 2.6, and in accordance with Gaberšek and Durran (2004), a $\hat{H}$-value where partially blocking of the airflow over a barrier may be expected.

Figure 11 shows wind speeds, wind directions, and power production in similar manners as in Fig. 8. The wind direction

at A1 and A3 vary mainly between 130° and 155° throughout the period considered. The wind at A2 has a slightly stronger southerly wind component, with directions varying between 145° and 180°. The wind is relatively weak at all clusters, with the A3 wind being generally weaker than the wind at A1 and A2. However, the A3-wind speeds occasionally exceed the wind speeds at the turbine clusters upstream. These wind speed peaks indicate that occasionally the kinetic energy becomes sufficient to allow for flow over the mountain and mountain waves to form. This observation does to some extent agree with the

variations in the non-dimensional mountain height in Fig. 10, although the temporal pattern does not align with the A3-wind speed peaks. The absence of a clear correlation between the $\hat{H}$ and the A3-wind speed pattern may, for instance, be attributed to the different temporal resolutions of the two datasets, with the hourly ERA5-data not providing the same level of temporal details as the 10-minute observational data from the wind park. Furthermore, the low wind speeds are reflected in the power production, with power production below 1 MW at all clusters. The power production is the lowest at A3, where the wind

speeds are below the cut-in threshold on several occasions. WRF is able to reproduce the observed wind and power production well between 09:00 and 15:00 LT. The simulated wind directions vary between 130° and 170°, and in agreement with the observations, the wind has a slightly more southerly component at A2 than the wind at the other two clusters. The simulated wind speeds correspond well with the observations, however, the periods with weaker winds at A3, compared to the upstream clusters, are shorter than in the observations, and the peaks, where the A3 wind speed exceeds the A1 and A2 wind speeds,

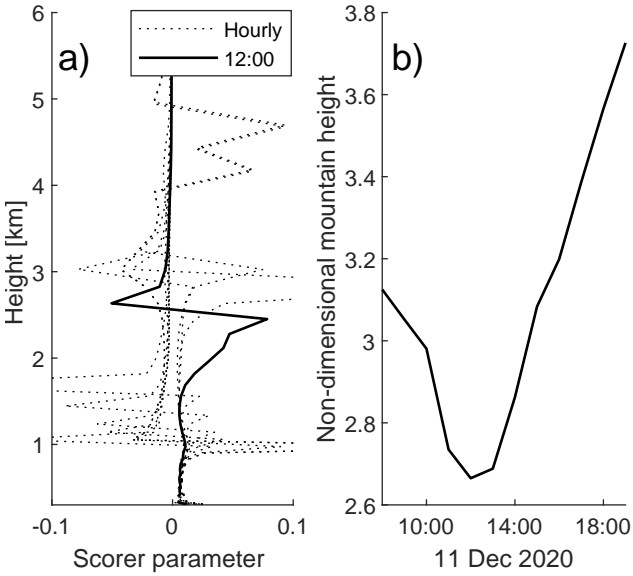

**Figure 10.** Case study 2 on the 11 December 2020: a) Hourly Scorer parameter (along x-axis) at altitudes indicated along the y-axis. b) The non-dimensional mountain height $\hat{H}$.

last longer. After about 15:00 LT, the agreement between the observation and the WRF simulations are reduced. The WRF simulations exhibit larger variations in the wind direction at A2 and A3 compared to the observations, and the wind weakens to below the cut-in threshold at all clusters.

Figure 12 is similar to Fig. 9, but represents the second case study on 11 December at 12:20 LT. The observed wind speeds in Fig. 12a) vary between 1 ms$^{-2}$ and 8 ms$^{-2}$. At wind park A, the winds are weak across the mountain top, and the lowest wind speeds, close to the cut-off threshold, are found on the mountain lee side. The turbines located on the mountain sides parallel with the airflow encounters the highest wind speeds. The WRF model is able to reproduce the spatial pattern of the observations well, with weak wind across the mountain top, considerably lower wind speeds on the mountain lee side, and the strongest winds on the western side of the wind park, as seen in Fig. 12b). The simulations do have slightly less spatial variations in comparison to the observations, with simulated wind speeds ranging from about 2 ms$^{-2}$ to 7 ms$^{-2}$.

Figure 12c) reveals large variations in wind speeds over the mountain areas, from 0 ms$^{-2}$ and up to 9 ms$^{-2}$ on the western side of the mountain of wind park A. The arrows indicate wind from the southeast as the flow climbs towards the mountain top. Just below the mountain top, the wind decelerates, and the arrows indicate that some of the airflow is diverted around the mountain. On the lee side of the mountain, a large blue area with wind speeds close to zero, indicates that the flow at 85 m agl is blocked by the mountain.

The vertical cross section in Fig. 12d) indicates high static stability from the surface and up to 3000 m asl. From the surface and up to about 2000 m asl, the tangential wind speed is decreasing with altitude. At 2000 m asl a critical layer is present with reversed wind speeds indicated by the red colour, in agreement with the Scorer parameter in Fig. 10. The airflow is



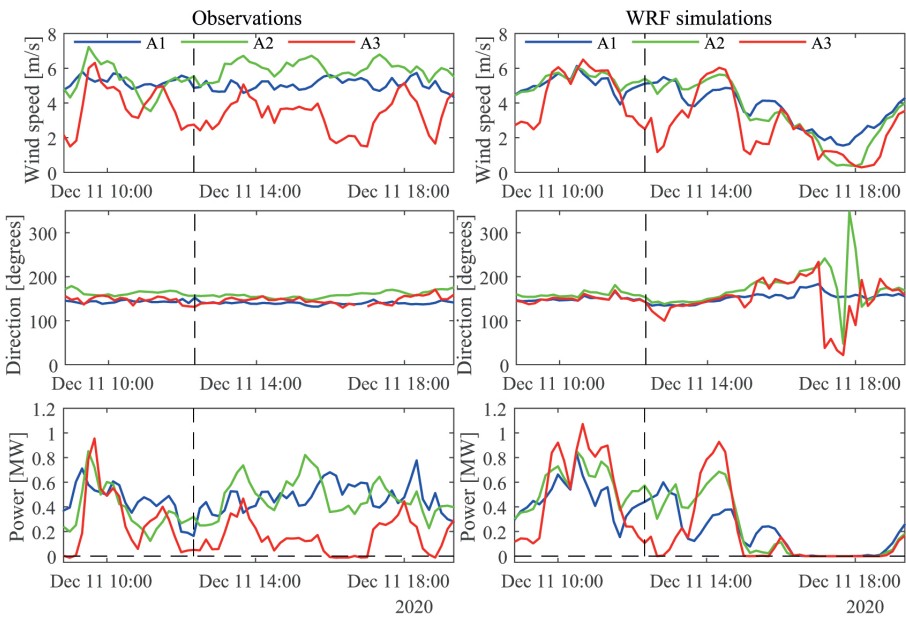

**Figure 11.** Observed (left) and simulated (right) wind speeds (top), wind directions (middle) and power production (bottom) for the three wind turbine clusters A1 (blue), A2 (green), and A3 (red) on 11 December 2020. The vertical dashed lines indicate the time of Fig. 12.

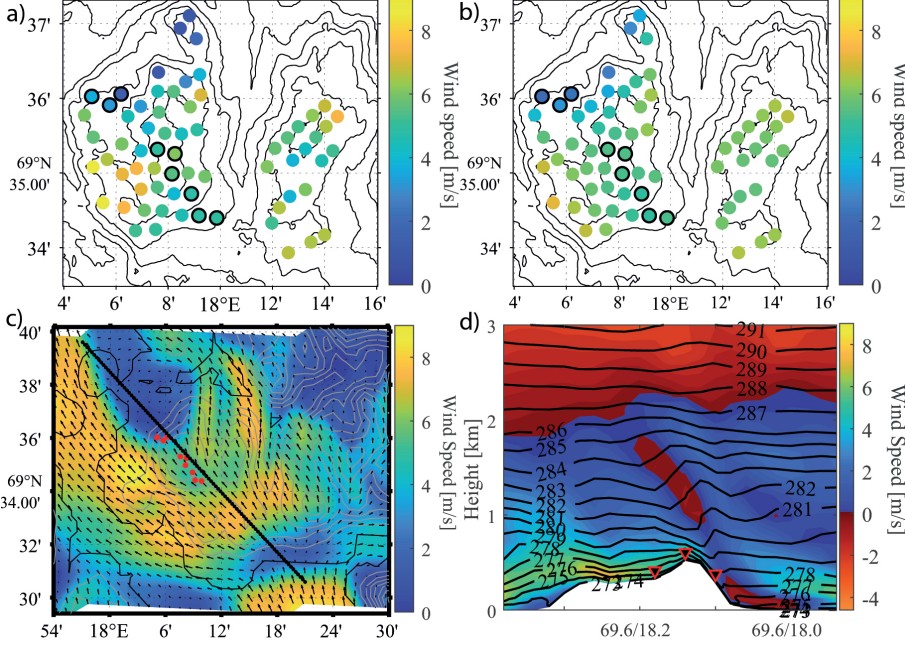

**Figure 12.** Similar to Fig. 9 for case study 2 on 11 December 2020 12:20 local time.





approaching the mountain with tangential wind speeds of 5-6 ms$^{-2}$ close to the surface and decelerates toward the mountain top. The isentropes show a small response to the mountain barrier, suggesting that the airflow is only partially blocked, and in agreement with the Scorer parameter in Fig. 10, the amplitude is growing and upstream tilting up to about 1500 m asl, and an area of reversed winds indicates breaking of the wave and a self-induced critical layer. However, the contour lines are only slightly compressed on the lee side of the mountain, and the downslope winds are not accelerated at the A3-area.

## 4   Conclusions

This study documents frequently occurring mountain-wave-induced accelerated downslope winds and their impact on the power production in two wind parks situated on two nearby mountains in Northern Norway. Wind park A and wind park B consists of an array of 67 wind turbines. For clarity and analytic simplicity, wind turbine clusters have been defined such that under SE-wind conditions, the clusters A1 and B1 are located upstream of the mountain crest, A2 and B2 at the crest, and A3 and B3 downstream of the mountain crest. The selected time periods studied consist of winds predominantly from the southeast and comprise 1104 hours of wind and power production data. During the selected time periods, the observed wind speeds at A3 are higher than the wind speeds at A2, 80 % of the time. Consequently, the power production of the A3-turbines, is 51% higher in comparison to the power production at the A1 cluster, and 19% higher in comparison to the power production at A2. Similar wind patterns and enhanced power production on the lee side of the mountain is also seen in wind park B.

The non-dimensional mountain height, $\hat{H}$, is a key parameter for describing the development of mountain waves, and in this study a strong relationship is found between $\hat{H}$ calculated from ERA5 data retrieved at an upstream location, and the wind speeds at A3. The results of this study suggest that the non-dimensional mountain height can be used to indicate whether enhanced power production should be expected at the A3-turbines, or the opposite, if blocking should be expected, with weaker winds and lower power production at A3 compared to the turbine cluster further upstream. Future studies should attempt to further strengthen the relationship between $\hat{H}$ and the A3 wind speeds by including meteorological data from multiple sources and upstream locations. In addition, future studies should investigate whether a similar relationship can be identified in wind parks elsewhere. One of the reasons for the strong relationship between $\hat{H}$ and the downslope wind speeds in wind park A, may be the long and well-defined fjord, channelling the wind rather undisturbed towards the mountain of the wind park. For other wind parks impacted by downslope windstorms, a more complex terrain upstream may lead to different results.

Finally, the WRF model reproduces the main mean wind features observed in the wind parks well, with higher mean wind speeds at the turbines located on the lee side of the mountains compared to the turbines located upstream. The simulated wind speed distributions at the three turbine clusters in wind park A, share the characteristics of the ones of the observations, with a higher frequency of wind speeds above 12 ms$^{-2}$ at A3 compared to the turbine clusters upstream. However, the difference in the wind speed distributions between the three turbine clusters are not as pronounced in the simulations compared to the observations. In addition, the deviations between the simulations and observations increase in line with the increasing complexity of the airflow across the mountain. The highest error is found at the mountain lee side, indicating that the model is not able to accurately reproduce the frequency or the strength of the accelerated downslope winds.



*Code availability.*  The WRF namelists are available for download at 10.5281/zenodo.15845751.

*Author contributions.*  KS conceptualized and administered the project; acquired resources; investigated, validated, visualised, and provided formal analysis; design of methodology; implementation of and model configuration;analysis software; writing of original draft, reviewing and editing. EMS conceptualized the project; design of methodology; analysing data; review and editing of draft. YB conceptualized the
505  project;supervision;design of methodology;acquired resources; review and editing of draft.

*Competing interests.*  The authors declare that they have no conflict of interest.

*Acknowledgements.*  This work is supported by Troms county and industry development fund under the project title, "Renewable energy in the Arctic – academy and business in a joint effort" RDA12/46. The authors would like to thank Nordlys Vind for the data from the wind parks. The simulations were performed on resources provided by Sigma2 - the National Infrastructure for High-Performance Computing and
510  Data Storage in Norway.



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
