# Peer review of "Mountain wave and downslope winds impact on wind power production"

_Wind Energy Science, 2025_

## Author Comment (AC1)

**Dear reviewer,**

We sincerely thank you for your time and effort in reviewing this manuscript. We appreciate your valuable feedback that has helped us improve the clarity and overall quality of the manuscript. Below you will find your original document. We have provided point-by-point responses directly beneath each of your **comments**, covering both major and minor points.

Sincerely,

Kine Solbakke, Eirik Mikal Samuelsen and Yngve Birkelund

Kine Solbakke, Eirik Mikal Samuelsen and Yngve Birkelund

Referee #2

**Introduction**

This manuscript examines hub-height wind speed and power production data from two wind farms located in moderately complex terrain in Norway, and documents the link between lee-side wind acceleration and environmental properties supportive of mountain wave propagation. Some evidence of the match between observations and simplified linear mountain wave theory is presented: for instance, lee-side acceleration is often observed when the dimensionless mountain height NH/U exceeds one. This is hardly a novelty from a mountain meteorology or mesoscale meteorology standpoint, but I might agree that the implications for wind energy harvesting went unnoticed so far. The manuscript also evaluates the ability of state-of-the-art hindcast simulations, performed with the WRF model at 750 m grid spacing, to resolve the lee-side acceleration. The model apparently captures the observed wind variability, but not entirely. For instance, lee-side acceleration in the simulations is less pronounced than in reality. The accuracy of numerical simulations is evaluated over a long hindcast run (about 4 months), and two day-long case studies are also presented.

**Recommendation**

The study contains a good initial review of literature on mountain waves (lines 27-48), which however contains some imprecise statements and misunderstandings, and lacks some fundamental concepts; accordingly, results based on the dimensionless mountain height NH/U and on the Scorer parameter are sometimes inappropriately discussed (see comments 1-8). The analysis of wind speeds and power production data is convincing, although some aspects could be improved (see comment 9). The global analysis of the accuracy of WRF simulations is thin (Figure 6b, Table 2), but the contents are seemingly solid. The purpose of the two case studies is not explained anywhere; they don't really add much to the scientific content; most importantly, some statements in the case study discussions are gross misinterpretations (see comments 8-9). Finally,

some technical aspects of the simulations are debatable (see comments 11-14). All considered, I would recommend requesting major revisions before considering publication.

**Major Comments**

1. **NH/U is an important quantity, but in a specific context: it serves as a nonlinearity parameter for linearly stratified flow with uniform wind speed (constant U and N). Yes, the critical value above which upstream blocking and leeside acceleration are expected is of order 1, and somewhat dependent on the geometry of the orography; but this concept only applies to flows that match reasonably well the constant N-constant U assumption. Real-world flows over mountains display extreme horizontal and vertical variability of wind speed, direction and stratification, so NH/U is generally not such a good predictor as one might expect**.

   Thank you for this valuable feedback on this point. Our approach is based on idealized models where N and U are constant with height, and we will add a section to the manuscript to clarify this for the readers. We will also clarify for the readers the method we have used to approximate U and N to a constant value with height. Despite limitations of the concept of the nondimensional mountain height, our result suggests that H may still be a useful indicator of the behavior of a flow encountering a mountain. The following section is added to our revised manuscript (line 168-179):

   *The theory of the non-dimensional mountain height was developed primarily for idealized flows, in which U and N are constant with height. Real-world flows will typically display vertical variability in both wind speed, wind direction and stratification, consequently, U and N must be approximated as constant with height(Reinecke and Durran, 2008). Despite the limitations of the concept, $\hat{H}$ is widely used to indicate real world flow behavior (e.g. Overland and Bond (1995); Jiang et al.(2005); Mobbs et al. (2005)). In this study the Brunt-Vaisala frequency is estimated by the following bulk method:*

   $$N = \sqrt{\frac{g}{\theta}\frac{\Delta\theta}{\Delta z}}$$

   *where $\theta$ is the potential temperature. It is assumed that the airflow that interacts directly with the mountain is spanning from the surface and up to a height of about 1 km, and $\Delta\theta$ and $\Delta z$ are therefore calculated between the lowest ERA5*

*model level (about 310 m asl) and model level number 120 (varying between 930-1000 m asl). The potential temperature $\theta$ in Eq. 3 and the tangential wind speed U in Eq. 2 are calculated using model level number 127. With heights varying between 553-586 m asl, the model level number 127 is the level closest to the real height of the mountain of wind park A. The mountain height h0 is set to be 550 m according to the real terrain within the A2-turbine cluster (Table 1).*

2. **Whenever a low-level inversion is present, concepts based on shallow-water flow are usually a lot more useful than internal gravity wave theory with constant N-constant U. Sometimes, downslope windstorms are caused by the transition from subcritical to supercritical conditions in "hydraulic" flow, capped by a strong inversion which acts as a density discontinuity. Here, sub- and super-critical refer to the shallow-water Froude number, Fr=U/sqrt(g'H). See for instance Durran (1990).**

We appreciate the reviewer's suggestion regarding the use of shallow-water flow concepts in the presence of a low-level inversion. Shallow-water models and hydraulic theory may indeed provide insight into the relationship between upstream weather conditions and the observed accelerated winds at the A3 wind turbines. For instance Mobbs2005, found a correlation between both, the Froude number (shallow water theory) and nondimensional mountain height (internal gravity wave theory). In our research, metrics based on the internal gravity wave theory have provided encouraging insights, and in future research we would suggest to include metrics based on shallow water theory as well.

In the introduction of our revised manuscript (line 48-51), we have included the following:

*"A third theory describing the mechanism of downslope windstorms is based on hydraulic theory and shallow water equations (Jackson et al., 2013). Downslope windstorms occur when subcritical upstream flow becomes supercritical over the mountain crest and accelerates down the lee slope. Further downstream, the energy is dissipated in a hydraulic jump where the subcritical conditions is restored."*

We have also added the following to Sect. 3.1 (line 287):

*"Future work should address additional metrics, such as the Froude number, developed based on hydraulic theory."*

3. **Although the definition of NH/U seems straightforward, this parameter is not easily evaluated. See Reinecke and Durran (2008).**

Thank you for this constructive comment, which has led to improvements in the manuscript. We have added the following paragraph to Sect. 3.1 in the revised manuscript (line 260-265).

*"Although the result of this study indicates a relationship between $\hat{H}$ and the downslope winds at A3, the results should be interpreted with caution. In particular, the approximation of a constant Brunt-Vaisala frequency is not straightforward, as demonstrated by Reinecke and Durran (2008). Reinecke and Durran (2008) applied two common approximations for N, referred to as the averaging method and the bulk method, to multiple cases with vertically nonuniform stability. Neither of the two methods provided a single best method to estimate a constant N. However, compared to the averaging method, the bulk method provided a better prediction of whether a flow will pass over the mountain or around it."*

4. **Another relevant aspect totally ignored in this manuscript is that, besides mountain height, mountain width matters too. In analogy to NH/U, it is possible to define a dimensionless mountain width NL/U, where L is the mountain half-width. Mountain wave response is known to be affected by NL/U as much as by NH/U (see for instance Sachsperger et al 2015). NL/U serves as a hydrostaticity parameter. By ignoring NL/U, this study implicitly assumes that waves always propagate nearly hydrostatically. This might not be true in general. Although, indeed, strongly stratified flow over a broad mountain is likely hydrostatic. The reason why NL/U and hydrostaticity are important is that, for sufficiently small NL/U, the maximum wind speed in cross-mountain flow occurs at mountain top, not on the lee slope.**

Thank you for this helpful suggestion. We have now calculated both the vertical aspect ratio and the hydrostatic parameter and added this information to our revised manuscript.

As the results shown in Sachperger et al. 2015, breaking of vertically propagating mountain waves typically occur when NH/U ≈ 1. Sachperger et al. 2015 shows that the formation of breaking mountain waves also depends on the vertical aspect ratio of the mountain. In our case, assuming the mountain of wind park A is about 12 km in SE-direction at the base, gives an L of about 6 km. With a height of 550 meters, the aspect ratio is about 0,09.

The hydrostatic parameter for all of the selected SE-events in our study, with the lowest value being about 4 (mean value about 19). This means that hydrostatic wave formation will be important in all SE-events.

We have added the following to our manuscript:

Method section line 180-184:

*"Additional non-dimensional parameters describing mountain wave characteristics are the vertical aspect ratio h0/L, and the hydrostaticity parameter NL/U where L is the half-width of the mountain. These are metrics describing whether the wave developing will be within the hydrostatic or non-hydrostatic regime. Mountain waves only exist when NL/U > 1. For NL/U >> 1 vertically propagating waves dominate with minimal wave motion downwind Jackson et al. (2013)."*

The results are presented in Sect. 31 line 245-252:

*"Sachsperger et al. (2016) found similar $\hat{H}$-values for wave breaking in idealised flow over an obstacle. However, at what $\hat{H}$-value the wave breaking first occurred also depended on the vertical aspect ratio. For aspect ratios of 0.1 and 0.05, wave breaking occurred when $\hat{H}$ = 1. For vertical aspect ratios outside of this range, wave breaking did not occur before $\hat{H}$ = 1.25 (Sachsperger et al., 2016). Along the SE-direction, the mountain of wind park A is approximately 12 km wide at sea level, corresponding to a half width of 6 km and a vertical aspect ratio of about 0.09. The hydrostaticity parameter, given by NL/U, is estimated to be above 4 for all SE-events, and hence within the hydrostatic regime where waves propagate nearly hydrostatically."*

Please take note of other changes in Sect. 3.1 in the revised manuscript. In response to the other reviewer's request, we have conducted additional sensitivity analysis. The results and the corresponding discussion have been added to this section.

5. **Similarly to NH/U, the Scorer parameter is an important parameter in a specific context. It is relevant for the description of wave trapping, i.e., horizontal propagation of resonant gravity waves within a wave duct. The Scorer parameter is really useful if one wants to show that variable stratification, wind shear, or wind curvature, lead to lee wave trapping; but trapped lee waves are not discussed anywhere in this manuscript! Herein, the Scorer parameter seems to be used only as an indicator of the presence of a mean-state critical level in the wind profile (U=0, or unbounded Scorer parameter; Fig. 7 and 10). However, wave breaking in the two case studies is visibly not caused by a mean-state critical level. All in all, there seems to be no good reason to use the Scorer parameter in this study. Note also that, if wind shear and curvature are neglected, the Scorer parameter (N/U) and the**

**dimensionless mountain height (NH/U) carry essentially the same information.**

We thank you for this comment. As the reviewer points out, we do not discuss lee waves, and case study 1 does not suggest wave breaking at a critical layer as the mechanism behind the accelerated downslope winds. We also agree that the Scorer parameter indicates conditions under which these wave phenomena could occur. However, the Scorer parameter also indicates whether upstream conditions occur that favor vertically propagating mountain waves and wave breaking.

When the conditions for decaying waves or trapped nonhydrostatic waves are not met, the waves are able to propagate vertically. For waves with long wavelength this may happen when the Scorer parameter is constant with height. In addition to the decreasing density with height, the vertical variations in the Scorer parameter may also modify the amplitude of the hydrostatic waves (Holton2004). When the cross-mountain wind is decreasing with height, the Scorer parameter increases with height. This allows the wave amplitude to grow and eventually break. In the case where the mean flow goes to zero, the Scorer goes to infinity , and the amplitude enhancement leads to wave breaking. This can happen, as the reviewer describes, at a critical level. But amplitude enhancement and wave breaking can also occur earlier, which is apparent in Case 1. E.g. Rögnvaldsson, et al. 2011 describes wave breaking when there is a reverse vertical windshear, which is also the case in the current study. The enhancement of the Scorer parameter shown in Figure 4 is in line with this. The Scorer parameter discussed in Case 1 and 2, as well as in relation to the mean reversed wind shear presented in Fig. 4,provides additional information about the upstream weather conditions. As the reviewer has already pointed out in comment 1 and 3, the evaluation of the non-dimensional mountain height is not straightforward, due to the fact that U and N must be approximated to constant values, while in reality they vary with height. The Scorer parameter does not require any such approximations as the parameter is developed within a framework that does not require U and N to be constant with height (Jackson2013).

6. **Lines 240-256 are rather speculative, and I would recommend reducing them. I think it is fine to say simply that real-world measurements are likely to deviate (even a lot) from expectations based on linear theory. If the environment is heterogeneous in terms of N and U, there's no reason why linear theory predictions should hold. For instance, a very stable surface layer leeward of the mountain (=strong vertical variability in N), might prevent a downslope windstorm from reaching the foothills.**

Thank you for this constructive comment. Following your suggestions and given the limitations of the non-dimensional mountain height, we have reduced our analysis. Specifically, we have removed Fig. 5b) and the lines 240-256 in the original manuscript.

In addition, we have added the following to our revised manuscript (line 284-286):

"*Although many of the events appear to be consistent with the theory, there are also several events that deviate. This is not unexpected as the concept of the non-dimensional height is developed based on linear flow with approximately uniform U and N. However, the apparent relationship between the non-dimensional mountain height and the A3 wind speeds is encouraging.*"

7. **The authors seem to interpret lee-side wind acceleration and blocking as distinct phenomena; instead, they are closely related. Dynamically, blocking is caused by a large pressure maximum upstream of the mountain, while lee-side acceleration is caused by a large pressure minimum downstream of it. In constant N-constant U flows, large pressure perturbations upstream and downstream of a mountain occur in the same conditions (NH/U>1). See for instance the introductory review by Serafin (2025).**

Thank you for pointing out this relation. Unfortunately, we have not been able to obtain access to Serafin (2025). Nevertheless, we have addressed this comment by drawing on other relevant literature that covers the topic, and we provide our response on that basis.

It is correct that these two phenomena are closely related, and they may occur separately or simultaneously under similar value of Nh/U. This is nicely summed up in Baines & Smith (1993). According to theory of uniformly stratified flow past a three-dimensional obstacle, as the non-dimensional mountain height is increasing, two regions of stagnation may occur. One stagnation region forms above and downstream of the mountain and is associated with wave breaking. The other stagnation region that may form is close to the surface on the upstream side of the mountain. The second stagnation region is associated with an adverse pressure gradient and reduced wind speeds. Below this stagnation point, the flow is diverted nearly horizontally around the mountain, resulting in a wake formation on the lee side with low wind speeds and eddy formation due to the upstream blocking. Which of those two stagnation points occur first, depends on the obstacle shape. Baines & Smith (1993) found that for broader obstacles, the stagnation first occurred for nondimensional mountain of 1.05, at a height equal to z = h/2. As Nh/U increases, the stagnation point moves towards the top of the mountain. It is suggested that the upstream stagnation point reduces the

effective height of the mountain and hence reduces the amplitude of the vertical wave. Full blocking, i.e. stagnation at a height equal to the height of the obstacle, occurred for very high values of Nh/U.  In our results we can see tendencies of reduction in the effective mountain height as the A3/A2 wind speed decreases as Nh/U increases above 1.5. Case study 2 suggests that the upstream stagnation point is still below the mountain top and suggests partial upstream blocking of the flow.

We have gone through the manuscript to make sure our language is more precise. We have also toned down the wording in places, rather than categorical suggesting upstream blocking, we now simply describes it as lower wind speeds at A3 compared to A2.  More substantial changes are made in for Case 2, in Sect. 3.4.2 in the revised manuscript.

We have also added the following to the introduction line 93- 96

"Combined with pre-development wind measurements, Ĥ would indicate whether winds are more likely to be diverted around the mountain, resulting in a wake formation on the lee side with low wind speeds and eddy formation due to the upstream blocking, or if the flow is more likely to pass over the mountain (Baines and Smith 1993)."

8. **Line 278 ("low wind speeds at A3 can be due to blocking of the air flow") and the whole discussion in Sec. 3.3.2: Orographic blocking occurs upstream of a mountain, not downstream of it. More likely reasons for low wind speeds on the lee side are the presence of a shallow and very stable surface layer (which would impede the penetration of downslope winds down to the level of A3 turbines); or wake effects such as in atmospheric rotors. Superficially, the spatial distribution of wind speed in Fig 12 is reminiscent of the flow structure often observed in Bora flow, with weak winds leeward of the highest mountains. See for instance Fig 11 in Gohm et al 2008 or Fig 12 in Gohm and Mayr 2005.**

Thank you for bringing more attention to case 2. We do agree that there might be several reasons that can explain the lower wind speed at A3 compared to A2 in this case.  The atmospheric rotors mentioned might be more likely than a very stable surface layer in our cases. The left figure below shows the observed temperature at A3 (red) and A2 (blue) during the study period, and the right figure shows the difference between the temperature at A3 and A2. The figures show that the temperature is always higher at A3 than at A2, indicating that there are no cases with low-level strong inversion at A3. In addition, the ocean is ice-free and typically warmer than the atmosphere in this area during the winter season.

We cannot say for sure which effect is causing the relative low wind speeds at A3, however, the wind speeds are generally low across the mountain in this event, and no enhanced wind speeds are observed at any of the turbines located at the mountain leeside. In addition, the wind turbines extract energy from the flow and may reduce the downslope wind acceleration.

[Figure]

In the revised manuscript we have added the following to the discussion of Fig. 13 (Fig. 12 in previous draft)  line 503-509  and line 518-524:

« ......*a large blue area with wind speeds close to zero. The weak winds in this area are interpreted to be a result of partial upstream blocking. Partial upstream blocking occur when a stagnation point develops on the windward side of the mountain and is described in Baines and Smith (1993). Below this point, the wind is diverted horizontally around the mountain, resulting in a lee-side wake with eddy formation. The air above the stagnation point will pass over the mountain, however, the efficient height of the mountain will be reduced. An alternative explanation is that the large blue area of weak winds results from atmospheric rotor formation downstream of a downslope windstorm (Mobbs et al., 2005). However, the absence of accelerated winds at the leeward turbines suggest otherwise..*"

« ......... *However, the contour lines are only slightly compressed on the lee side of the mountain, and the A3 winds are weaker than the winds at A2. As noted above, one possible explanation of the weak winds at A3 may be partial upstream blocking, another less likely reason is the presence of atmospheric rotors. Another explanation may be that a shallow and very stable surface layer prevents the downslope winds from reaching A3. However, the observed temperatures at the A3 turbines are consistently higher than at A2 throughout the entire study period (not shown), suggesting the absence of low-level inversions. In addition, a low-level inversion is also unlikely given the ice-free oceans are typically warmer than the atmosphere in the winter season. Furthermore, energy extracted from*

*the airflow by the turbines may reduce any potential accelerations of the downslope winds. »*

9. **Please describe the rationale for the choice of the case studies. It is really not clear. It looks like the dynamics are understandable based on simple theory in case 1, while they are not in case 2.**

   We appreciate the reviewer's comments and acknowledge that the reasoning behind the two case studies was not sufficiently explained in the manuscript. We have included case 1 to show the typical case with high wind speeds and stronger winds at A3 compared to A2. This represents the main situation at the wind park during SE events. In addition, we have included case 2 where we have the less frequent opposite situation with lower wind speeds in general and a lower wind at A3 compared to A2. We have added the following to the manuscript (line 420-424):

   "The following analysis investigates two different events: Case 1 represents a typical SE-event, with stronger winds at A3 compared to A2. Case 2 represent the less frequent event, with weaker wind at A3 compared to A2. The analysis is motivated by the observations showing that under comparable wind directions, the wind speeds and power production at A3 relative to A2, can vary substantially."

10. **Figure 5 represents only data points from SE wind events. One might still argue that wind speeds at A3 could be higher than at A2 or A1 for reasons unrelated to leeside acceleration; so the normalized wind speeds A3/A2 and A3/A1 could be >1 even if the wind at P blows from directions other than SE. Does the distribution of normalized wind speed look substantially different for other directions? Could one check similar ratios for NW flow? I think the information could be easily added to this Figure.**

Thank you for this constructive comment. We do agree that it would be interesting to evaluate if there exists a similar relationship when the wind comes from the NW. However, the main wind direction in the wind park is from the SE (Solbakken et al. 2021 Figure 1). The SE wind direction is especially pronounced during the winter months, and because this study covers the period from September to January, the NW wind direction (300° -345°) is observed for less than 5% of the study period.

To clarify, all wind speeds presented in Figure 5 are times selected based on the wind direction within the wind park, i.e. every green dot corresponds to a time stamp when the observed wind direction at 40 turbines or more are within the selected SE-sector.

During the selected SE-events, the ERA5 wind direction at location P may in fact not be from the SE. However, and not unexpected, the wind direction at P is typically within the SE-sector during the events studied.

At this high latitude location, the sun is either low or below the horizon during the study period, so thermally driven winds are unlikely to develop. In addition, due to low incoming radiation during the winter months, flow from SE will typically be stably stratified. Air flow from NW during winter are less likely to be stably stratified due to the relative high sea temperature along the coast with relatively colder air above leading to unstable conditions.

The terrain in the wind park is gently undulating with rocks and very low to no vegetation. There are no larger trees. In addition, to reduce the impact on small local terrain effects, we have taken the mean of three turbines within each cluster. We therefore do not think that local friction effects are the reason behind the lower wind speed at A2 than A3.

The left figure below shows the A3/A2-wind speed and the wind direction (at A2). The green dots represent the wind from SE, and the red dots represent the wind from the opposite direction. As can be seen in the figure, during the study period the wind comes substantially more often from SE (37%) than from NW (5%). The wind from SE is typically higher (74%) at A3 than at A2 (24%), while from NW, the wind is typically lower at A3 (63%). The figure to the right is a similar figure, however for the A1/A2 wind speed. As can be seen, when the wind flows from both NW and SE, the wind at A2 is typically higher than at A1.

[Figure]

11. **Line 137: ERA5 profiles at point P. Judging from Fig. 3, point P is well within the highest-resolution WRF simulation domain D03. If the purpose is to evaluate the ability of WRF to reproduce downslope windstorms and their impact on power production, I really do not understand the reason why the**

**properties of the upstream environment are drawn from ERA5 instead of WRF. Leeside flow properties in WRF simulations should be more closely linked to upstream profiles from the same simulations, than to ERA5 upstream profiles (a different model, with much coarser spatial resolution).**

As described in the manuscript, the relationship between the observed A3 wind speed and the upstream weather conditions, presented in Fig. 5, is based on the ERA5 reanalysis. One reason for using the ERA5 data at location P, with a low resolution, as opposed to the high-resolution WRF simulations, is that the ERA5 data provides a mean state of the upstream weather conditions free of local terrain effects. In reality, the complex terrain may modify the weather conditions at location P in such a way that it is not always representative of the upstream weather conditions of the wind park. This will also be the case if location P is positioned closer to the wind park. The high-resolution WRF simulations will be able to capture some of these smaller scales local effects and hence differ from the larger scale mean weather conditions. We could also have used the D01 or D02 WRF simulations, however for the purpose of this study, we do think it is important to minimize any small-scale-terrain induced disturbances that may be present also in these domains. This is addressed in the manuscript line 255-257,

> "One reason the results appear to agree reasonably well with the theory may be that the lower-resolution ERA5 data, as opposed to local observations or high-resolution WRF simulations, provide a mean state of the atmosphere free of local terrain effects at location P."

In addition, this study has three objectives:

> 1) To evaluate the relationship between the upstream weather conditions and the accelerated wind speeds at A3. This is done by use of observations within the wind park, and ERA5 data extracted at location P. (Sect. 3.1)

> 2) To document the presence of mountain waves/downslope windstorms and how it impacts the wind parks in the study (Sect. 3.2)

> 3) To evaluate the ability of the WRF model to reproduce the observed wind patterns in the wind park during SE-winds. (Sect 3.3)

The evaluation of the relationship between the $\hat{H}$, based on ERA5, and the normalized A3 wind speeds is not meant to contribute to the discussion of the ability of the WRF model to reproduce downslope windstorms.

Rather, the purpose of the evaluation of the relationship between $\hat{H}$ and the A3-wind speed is to investigate if $\hat{H}$ can be a useful parameter both for planning purposes for a wind park and for power prediction purposes. ERA5 represents a readily available dataset that to a low cost can provide wind park developers with upstream weather information.

To make our objective and motivation clearer to the readers, we have added the following text to the introduction of the revised manuscript (line 92-96):

*"This study evaluates whether a similar relationship exists between $\hat{H}$ and the accelerated downslope winds observed within the wind park. If such a relationship exists, the non-dimensional metric could be valuable to the wind-energy community. Combined with pre-development wind measurements, $\hat{H}$ would indicate whether winds are more likely to be diverted around the mountain, resulting in a wake formation on the lee side with low wind speeds and eddy formation due to the upstream blocking, or if the flow is more likely to pass over the mountain (Baines and Smith (1993)). In the operation phase, $\hat{H}$ could indicate whether to expect enhanced power production on the mountain leeside."*

12. **Judging from Fig. 1, the area of the two wind parks is about 10 km across. At 750 m resolution, this means about 13 grid points. This is marginally adequate to resolve atmospheric variability within the area, especially considering that the effective resolution of a NWP model is usually at least 7dx. This means that all variability at shorter scales is severely damped by the dynamical core of the model, in order to preserve numerical stability (see e.g. Skamarock 2004). This likely explains why the simulations presented here do not resolve much the of observed variability in the wind.**

We appreciate the observations made by the reviewer and can confirm that the model is configured with a horizontal resolution of 700m, corresponding to an efficient resolution of ~4.9 km. We acknowledge that a higher horizontal resolution could have been beneficial for the results. However, the Fitch wind farm parameterization scheme is recommended to be used in combination with a horizontal resolution no smaller than 5 rotor diameters. In our case the 4.2 MW turbines have a rotor length of 130 m, and hence the grid size should be no

smaller than 650 m. We have addressed this point in our revised manuscript, and line 393-400 now includes the following sentences:

*"For the purpose of the current study, the horizontal resolution is carefully selected to balance the benefits of higher details with the constraints imposed by the Fitch wind farm parameterization, which recommends a grid size of at least 5 rotor diameters (Fitch et al., 2012). For this particular study, given that the mountains of the wind park are approximately 10 km across, a 700 m grid resolution, corresponding to an effective resolution of $7 \times \Delta x \approx 4.9$ km, may not adequately resolve the mountain-induced flow perturbations. Perturbations, such as mountain waves, on scales smaller than the effective resolution are damped, and likely contributing to the model not being able to reproduce some of the observed variability in the wind (Skamarock, 2004). Future studies should evaluate the potential benefits of a finer grid resolution, though this would require reconsideration of the turbine representation."*

13. **The WRF orography is drawn from the GMTED digital elevation model. The maximum resolution of this DEM is 7.5 arc seconds (about 230 m), but I think the version available in the WRF pre-processing system is at 30 arc seconds (about 1 km). Again, this is marginally adequate.**

Thank you for pointing this out. We confirm that the orography is drawn from the GMTED digital elevation model with 30 arc second resolution. We have revised the manuscript (line 390-392) to address this point:

*"In addition, the terrain model used to configure WRF has a resolution of 30-arcseconds and is a part of the default setup of the WRF model. However, higher-resolution terrain models are available and could potentially improve the terrain representation in D03 domain."*

14. **Lines 182-184 explain that the hindcast simulations are actually 8 days long, and are connected by some kind of "interpolation". What kind of interpolation is alluded to here? Is it a weighted average between the two simulations, with weights changing over time? Please specify better. I understand that the authors do this in order to ensure that the WRF simulations do not drift away from the ERA5 analysis fields over a 4-month long simulation. However, I have never seen a solution based on interpolation before. Weighted averages of different weather forecasts are notoriously rather unphysical. The state-of-the art technique to prevent drifting is spectral nudging (see e.g. Waldron et al 1996 and Liu et al 2012). I personally find this a severe shortcoming in experiment design, but I**

**concede that it only affects about 7% of simulated period (12 hours every week).**

After reading your comment we realize that our method is not clearly described in the text. The linearization you refer to in your comment is a part of the post-processing and does not affect the simulations. We have included the following sentences to clarify this in the revised manuscript:

*"The ERA5 global reanalysis is used as initial and boundary conditions for the simulations. The simulations are run for 8 days, with the first 12 hours considered spin-up time. The next 12 hours of the simulation period has been interpolated with the last 12 hours in the previous simulation, to allow for smooth overlap of the time series. The simulations cover the period from 4 September 2020 to 24 January 2021. The wind data are retrieved from 85 m agl by vertical interpolation of the model levels, and from the given turbine locations by horizontal bilinear interpolation between the grid points. Similar to the observations, the cluster wind values, and power production are the mean values of each parameter at the turbines within each cluster."*

**Minor comments**

1. **Lines 34-35: Speaking of "fast propagating lee waves" is potentially very misleading. Mountain waves and lee wave fronts propagate opposite to the mean flow, and therefore they are generally stationary. Fast variations in local atmospheric properties near mountain or lee waves are most often not caused by wave "propagation", but by nonlinear wave interactions (Nance and Durran 1998), or by the transient phases (wave onset/demise) in non-stationary flow (Nance and Durran 1997, Grubisic et al 2015).**

   Thank you for this observation, we have removed this sentence from the manuscript.

2. **Line 41: More precisely, locally reversed air flow near the surface is typical of atmospheric rotors. These may or may not be connected with a hydraulic jump (e.g. Hertenstein and Kuettner 2015). Convective overturning and reversed flow connected with a hydraulic jump typically do not occur near the surface.**

   Thank you for the suggestion. We have revised the manuscript to include rotors to our description of downslope windstorms.

*"In addition, the accelerated downslope winds can terminate in a hydraulic jump accompanied by rotor development, further reinforcing the turbulence and potentially lead to locally reversed air flow downstream of the lee side of the mountain (Doyle et al., 2000; Doyle and Durran, 2007; Gaberšek and Durran, 2004)."*

3. **Line 60: This is strange wording. Static stability is a physical property of the atmosphere, so "various static atmospheric stabilities" sounds just as strange as "various atmospheric pressures".**

Thank you, and we agree. We have revised the manuscript and changed the text to: *"Radunz2021 studied how a range of static-stability conditions affected power production at two wind farms in Brazil situated on a plateau."*

4. **Line 95: "Straits" instead of "straights"?**

Thank you, this has now been corrected.

5. **Line 98, and elsewhere: please add a blank space between "m" and "s"; "ms" means milliseconds.**

Thank you for making us aware of this error. It has been corrected in the revised manuscript.

6. **Line 117: "Winds from SE are considered". I presume this refers to hub-height winds. Wind direction changes with height, and the manuscript makes abundant use of concepts form 2D (x-z) mountain wave theory; so it would be important to make sure that the flow is reasonably 2D, that is, without large directional shear. Is this feasible, with the data at hand?**

This is correct, and we have added "hub-height" to the sentence you are referring to. Unfortunately, we only have hub height wind and are unable to say anything about the wind shear/directional shear.

7. **Line 166: Horizontal resolutions of 10.5 km, 3.5 km, and 750 m. Is this correct? Nesting ratios in WRF must be integer numbers, preferably odd integers. 10.5/3.5 = 3, which is fine. But 3500/750 is 4.6666, which is not fine. Could you please verify?**

Thank you for making us aware of this error. The correct grid size for domain DO3 is 700 m. This has now been corrected in the revised manuscript.

8. **Line 204: "conditions in which both mountain waves might grow and break, as well as being dissipated at a critical level". I don't understand this distinction. Mountain waves "grow" and "break" even at a critical level. Dissipation (deposition of the wave kinetic energy into the mean flow) is a consequence of breaking.**

The have revised the wording to make the text more precise:

*"This distinct signature in the upstream large scale wind profile is quite striking and indicates upstream weather conditions in which mountain waves may either grow sufficiently to break and form a so called "self-induced" critical level or propagate vertically until it breaks at the mean-state critical level."*

9. **Line 272: "considerably" instead of "considerable".**

   Fixed.

10. **Line 346: "Several factors may impact the accuracy of the model simulations". I find it awkward to begin this discussion with an impact of microphysics parameterizations. Yes, microphysical schemes might be relevant (especially if the flow is often saturated), but they are certainly not the primary factor that comes to mind. Besides the aforementioned marginally adequate resolution of the numerical grid, I report two more factors that are most likely a lot more relevant than microphysics: 1) Surface friction parameterization, e.g. Richard et al 1989; 2) Predictability; small changes in upstream conditions can cause very large deviations in leeside response, especially in nonlinear flow regimes e.g., Reinecke and Durran 2009.**

    Thank you for this feedback. Based on your comment we have made several changes and additions to the discussion regarding the reasons why WRF is not able to more accurately reproduce the accelerated downslope winds. We have changed the wording in the line you refer to (379-395 in revised manuscript). Followed by a discussion regarding horizontal resolution, effective resolution and wind farm parameterization.

    *"Several factors may impact the accuracy of the model simulations. For instance, Reinecke and Durran (2009) found that small variations in the initial conditions led to substantial differences between forecast ensembles of downslope windstorms, including qualitatively differences in the characteristics of the upper-level wave breaking, as well as in the strength of the downslope winds. The accuracy of numerical simulations of downslope winds are also highly sensitive to the accuracy of the roughness length, land use and surface friction parameterization (Shestakova, 2021; Reinecke and Durran, 2009; Sachsperger et al., 2016)). While Rognvaldsson et al. (2011) highlights the importance of the micro-physical processes in the formation of downslope windstorms."*

11. **Lines 404-421: A lot of text in this paragraph describes the graphical elements of Figure 9, and should therefore be reported in the figure caption.**

Thank you for this comment. We have now revised the section of Case 1 and moved the description of graphical elements to the caption of Fig. 9.

12. **Line 420: "This is in accordance with the Scorer parameter in point P". It is really an excellent match with theory, but the description doesn't really explain why and could be more precise. Citing from Serafin 2025: "If N and U are nearly uniform and NH=U is supercritical, wave-breaking tends to occur at altitudes between 1/2 and 3/4 of the vertical wavelength (2piU/N in the hydrostatic limit)". Neglecting the curvature term in the Scorer parameter, I roughly assume N/U=3*10^-3 (Figure 7a). This maps to a vertical wavelength of 2100 m. Therefore, wave-breaking is expected between 1000 and 1500 m, which matches Fig 9d very well.**

Thank you for this perspective. It is interesting to see that hydraulic theory also matches the height of the wave breaking in case 1. the flow characteristics in case 1. The sentence "This is in accordance with the Scorer parameter in point P" refers to that the height of the breaking coincides with the height ranges in Fig 8a) (in the revised manuscript) where the Scorer parameter is constant and increasing between 1000-2500 m, suggesting conditions in which vertically mountain waves may grow and break.

We have included   the following in our revised manuscript:

*"This is in accordance with the Scorer parameter in point P. As shown in Fig. 8, the parameter indicates conditions that favor vertically propagating mountain waves and wave breaking between 1000-2500 m asl."*

13. **Line 484: "Strong relationship". I presume this statement alludes to Figure 5. It does not look like a "strong relationship". The Pearson correlation coefficient of the relationships between A3/A2, or A3/A1, and NH/U is not displayed, but is likely quite low (for good reasons, as extensively explained above).**

Thank you for this comment. We acknowledge that "a strong relationship" may be a bit overstated and have therefore reduced the wording to "a relationship". The paragraph in the conclusion (line 537-539) now starts with the following:

*"The non-dimensional mountain height, $\hat{H}$, is a key parameter for describing the development of mountain waves. By comparing $\hat{H}$  calculated from ERA5 data retrieved at an upstream location, and normalised wind speeds at A3, a relationship is found that agrees surprisingly well with the theory of the non-dimensional height."*

Regarding the Pearson correlation coefficients, you are correct, the coefficient would likely be low, as it describes linear relationships. In this case we do not

expect to find a linear relationship since for $\hat{H} \ll 1$ the normalized wind speed is expected to be close to 1, increasing above 1 for $\hat{H} \sim 1$, before decreasing when $\hat{H}$ increases further. We have added a version of Fig. 5 where we have included basic cubic fitting in Matlab to more clearly show this relationship:

[Figure]

14. Figure 1: Please add a latitude-longitude grid, to make it more comparable with Fig. 2 and 3

   Figure 1 is updated with an inserted map in the lower left corner indicating which turbines are A and which are B. In addition scales that indicate the actual distance are included in both the large map, and the inserted map.

15. Figure 2: Please add a bar scale to the map, so that actual distances can be clearly understood. I would also recommend adding a second panel with the resolved model orography.

   We have solved this with updating Figure 1 with bar scales and additional close up map of the wind park inserted into the figure.

16. **Figure 4: I would recommend showing also the variability of the profile, e.g. mean plus-minus standard deviation, or horizontal whiskers between 10th and 90th percentile. I presume the variability will be quite large.**

   We appreciate the reviewer's suggestion. The profile in Fig. 4 is indeed the mean, and as expected, there will be large variations in the profile between each event.

   We have added the following text in line 226: "It is expected that the vertical wind profile will vary between the different events."

**References**

Durran 1990: https://doi.org/10.1007/978-1-935704-25-6_4.

Gohm and Mayr 2005: https://doi.org/10.1256/qj.04.82

Gohm et al 2008: http://doi.wiley.com/10.1002/qj.206

Grubisic et al 2015: https://doi.org/10.1175/JAS-D-14-0381.1

Hertenstein and Kuettner 2015: https://doi.org/10.1111/j.1600-0870.2005.00099.x

Liu et al 2012: https://doi.org/10.5194/acp-12-3601-2012

Nance and Durran 1997: https://doi.org/10.1175/1520-0469(1997)054%3C2275:AMSONT%3E2.0.CO;2

Nance and Durran 1998: https://doi.org/10.1175/1520-0469(1998)055%3C1429:AMSONT%3E2.0.CO;2

Reinecke and Durran 2008: https://doi.org/10.1175/2007JAS2100.1

Reinecke and Durran 2009: https://doi.org/10.1175/2009JAS3023.1

Richard et al 1989 https://doi.org/10.1175/1520-0450(1989)028%3C0241:TROSFI%3E2.0.CO;2

Sachsperger et al 2015: https://doi.org/10.1002/qj.2746

Serafin 2025: https://doi.org/10.1016/B978-0-323-96026-7.00123-5

Skamarock 2004, https://doi.org/10.1175/MWR2830.1

Waldron et al 1996: https://doi.org/10.1175/1520-0493(1996)124%3C0529:soasfa%3E2.0.co;2

---

## Author Comment (AC2)

**Dear reviewer,**

We sincerely thank you for your time and effort in reviewing this manuscript. We appreciate your valuable feedback that has helped us improve the clarity and overall quality of the manuscript. Below you will find your original document. We have provided point-by-point responses directly beneath each of your comments, covering both major and minor points.

Sincerely,

Kine Solbakke, Eirik Mikal Samuelsen and Yngve Birkelund

Referee #1

**General considerations**

In this paper, the authors elaborate on the impact of downslope wind storms on wind speed at hub height and power production downwind of a hill or small mountain of some 550 m height. The results are based on two (close) wind parks with a total of 67 turbines in northern Norway. From a mountain wave perspective, this is not entirely new (and also not intended to be by the authors) but from a wind power perspective, this additional aspect for site selection certainly will add added value. The problem with the paper is, that the authors do not have 'good' data (the nacelle wind speed is certainly good for operational purposes, but of course constitutes a perturbed measurement per se (one places the instrument into the perturbation that one wants to observe…). So, basically the analysis has to rely on the modeling, the essential features of which are hard to validate (what really counts is the upwind stability (no observations available), the Scorer parameter as a function of height, the upwind topography (for different flow situations), i.e., the compatibility of the flow configuration with theoretical framework, of mountain waves. So, when relying on the model simulations (or having to rely) it would be desirable to see some more sensitivity analysis rather than demonstration of the occurrence at this particular site.

I have added some suggestions (major comments 1-3) how to possibly enhance the value of the existing simulations and also a major comment on which sensitivities could possibly be explored in more detail (major comment 4). All together, since there are

quite numerous detailed comments and one or the other major comment needs to be properly addressed, I call this 'major revisions required'.

Major comment

1. **The SE events are selected (which is fine in principle). The WD sector, however, consists of a range (from about 165 to 140 degrees) where indeed the approach flow rises from sea level to the 550 m high 'hill top' (so, non-dimensional mountain height is certainly appropriate), while for the WD range 140-120 degrees, the flow is in fact descending from a much higher mountain (cluster of peaks) to zero (only a few km horizontal distance), then rising over an even higher area. From idealized (e.g.doi:10.3390/atmos8010013 ) (but also real) studies we know that in this situation wave interference may play a crucial role. Did the authors consider a distinction according to wind direction within the SE sector (I perfectly realize that wind direction variability may not allow for such a fine distinction – but maybe a tendency will be visible)?**

We greatly appreciate this constructive feedback. We have now distinguished between the two sectors (120-140 degrees and 140-165 degrees) as you suggest, and the results are summed up in the figure below. The figures show a difference between the two sectors, with a better agreement with the theory for the sector 140-165 degrees (blue) than sector 120-140 degrees (red).

We have added the figure and the following paragraph describing the sensitivity to the selection of wind direction sectors to Sect. 3.1 in our revised manuscript:

*"The sensitivity to upstream geography is tested by dividing the SE-wind sector into two smaller sectors. In the first sector (120 ∘ - 140 ∘) the nearby upstream terrain is characterized by several high mountains (Fig.1). In the second sector (140 ∘-165 ∘), the air flow approaches the wind parks rather undisturbed through a long fjord. The distinction between the two sectors is made based on observed wind directions from one of the A2-turbines. Figure 6a) shows the results for the first sector, and Fig. 6b) for the second sector. Within the first sector, there is a tendency of increasing normalized wind speeds as $\hat{H}$ increases towards one, however, similar to Fig.5, there are some deviating wind speed values for $\hat{H}$ ~1.Within the second wind sector the relationship between $\hat{H}$ and the normalised A3 wind speed appears to agree better with the theory, with less spread in the normalised wind speeds when $\hat{H}$ ~1, as well as no normalised wind speed values below unity for $\hat{H}$ < 1.6. Figure 6b) supports our view that one of the reasons why the rather simple theory of the non-dimensional mountain height appears to some extent hold in this real-world case, as opposed to an idealised model, may*

*be the long and well-defined fjord that channels the wind from location P towards the mountain."*

[Figure]

2. **I wonder whether the argument could not be turned around, by also selecting a number of NW flows (the upwind ERA5 grid point is also available, for the Scorer parameter and non-dimensional mountain height. Wouldn't then A3 be the upwind and A1 the downwind site?**

Thank you for this comment. In addition to our answer here, we would also like to refer the reviewer to our answer to the second reviewer (major comment number 10).

We agree that applying the non-dimensional mountain height framework to NW-winds would be interesting. As you noted, for NW flow, A3 is upwind and A1 downwind of the mountain top. Mountain waves and downslope winds may also occur under NW flows. In this scenario, the A1 wind turbines may encounter higher wind speeds and higher power production than the A3 and A2 wind turbines. However, this wind direction is not considered in this study for two main reasons:

a) The main wind direction is from SE (Solbakken et al. 2021 Figure 1). From a wind power production perspective, the SE wind direction is therefore the most relevant direction. In addition, the SE wind direction is especially pronounced during the winter months, and because this study covers the period from September to January, the NW wind direction (300° -345°) is observed for less than 5% of the study period and the wind is typically weak.

b) Stably stratified flow is a prerequisite for mountain waves to form. During the winter months mountain waves are mainly expected to form when stably stratified air approaches the coast from the east and southeast. Wind approaching the coast from the sea during winter are less likely to be stably stratified due to the relative high sea temperature along the coast with relatively colder air above leading to unstable conditions, hence

mountain waves are less likely to form when the airflow is from NW or these are formed in a stable layer far above the mountain top not influencing the wind pattern at the wind park in a similar manner.

The left figure below shows the A3/A2-wind speed and the wind direction (at A2). The green dots represent the wind from SE, and the red dots represent the wind from the opposite direction. As can be seen in the figure, during the study period the wind comes substantially more often from SE (37%) than from NW (5%). The wind from SE is typically higher (74%) at A3 than at A2, while from NW, the wind is typically lower at A3 (63%). The figure to the right is similar figure, however for the A1/A2 wind speed. As can be seen, when the wind flows from both NW and SE, the wind at A2 is typically higher than at A1.

[Figure]

3. **The case studies. These are presented and discussed in a way to demonstrate their point (which, by the way, is not so clearly worked out, especially for case study two…. Is the intention to demonstrate that 'it is complicated'?). However, case study 1, for example, could possibly be used to elucidate some wind direction sensitivity. At about 2 pm on the 25th (Fig. 8), the observed WD abruptly changes, but does only go to about 150 degrees, while the modelled WD also changes but reaches some 130-140 degrees). Thus, the observed flow seems to clearly come through the fjord, while the modeled flow, at least for some 3 hours, seems to be modified by the high mountains possibly interacting (also with wave activity). Maybe another cross-section (as Fig. 8d) could help to understand some of the flow behavior.**

We appreciate the reviewer's comments and acknowledge that the reasoning behind the two case studies was not sufficiently explained in the manuscript. We have included case 1 to show the typical case with high wind speeds and stronger winds at A3 compared to A2. This represents the main situation at the wind park during SE events. In addition, we have included case 2 where we have

the less frequent opposite situation with lower wind speeds in general and a lower wind at A3 compared to A2. We have added the following to the manuscript line 420-424:

"The following analysis investigates two different events: Case 1 represents a typical SE-event, with stronger winds at A3 compared to A2. Case 2 represents the less frequent event, with weaker wind at A3 compared to A2. The analysis is motivated by the observations showing that under comparable wind directions, the wind speeds and power production at A3 relative to A2, can vary substantially."

Regarding figure 8 and the abrupt change in wind directions. As the reviewer points out, the observed wind direction at A3 changes abruptly from 160 to about 145 degree at about 2:40 pm. At the same time a similar abrupt change in wind direction at A3 and A2 is seen in the simulations, changing from about 166 to 140. After this the observed wind direction at A3 remains below 150 degrees, while the simulated A2 and A3 wind directions (from about 4.40 pm) increase back to above 160 degrees.

While the observed wind speed remains relatively stable despite the abrupt change in wind direction, there is a large increase in wind speeds from less than 10 m/s to close to 20 m/s at all three turbine clusters. The increase in simulated wind speed coincides with the change in wind direction to 140 degrees, suggesting that the approaching simulated wind is affected by upstream wind conditions, and perhaps upstream wave formation. As expected, the cross section taken at 3.40 pm differ substantially from the cross section in Figure 11 (in the revised manuscript) at 5 am the same day. At 3.40 the upstream wind speed is substantially stronger, however, there are no indications of wave breaking leading to accelerated winds at the leeside.

Although this is an interesting discussion, we do not think the difference between the cross section below, and the one in Fig. 9 in the revised manuscript are that different that it adds any new information to case study 1. A future study, however, should compare accuracy of simulations of events where the wind direction is within the sector 120-140 degree, and the sector 140-165 degree.

[Figure]

4. **Sensitivity studies: more than the question, whether downslope windstorms occur (and hence must have an impact on with energy and power production), it will b of interest, how this can appropriately be modelled with a model like WRF. The authors have made a number of choices: 1) location of P1 (or: determine the background flow characteristics from an ERA5 grid point (and if so, which), or from an average of WRF grid points (and if so, which); 2) levels from which N is diagnosed; 3) depth of what I assume is meant to be the boundary layer; 4) neglection of the non-linear term in determining the Scorer parameter; 5) definition of the variability across the chosen WD sector. It is not that I would doubt the authors' actual choices (they are at least not unreasonable), but there would be considerable added value from an sensitivity analysis (how sensitive are the results on this or the other choice?).**

We have revised our manuscript (line 265-272) to make it clear that we have performed some sensitivity tests and that the results appear to be only weakly sensitive to the selection of vertical layers. In addition we have performed a sensitivity test for wind sectors as a response to comment number 1 above.

*"To account for the uncertainty in estimating N and U, sensitivity analyses are performed. As described in Sect. 2.2, N is estimated using the bulk method in Eq. 2 between the ERA5 model levels spanning from the surface and up to a height of about 1 km. The sensitivity in the approximation of N is tested by changing the upper level to heights of about 855 m asl, and 1172 m asl. The results (not shown) appear to be only weakly sensitive to the choice of the upper vertical level used to calculate N, with the overall findings remaining consistent. U in Eq. 2 are approximated from the single ERA5 level closest to the mountain peak (about 580 m asl). The sensitivity to the approximation is evaluated by calculating U at*

*heights slightly below (500 m asl) and above (677 m asl) mountain top level. The results (not shown) are only weakly sensitive to the choice of model level."*

In addition to the sensitivity testing of different selection of vertical levels presented below. Changes in h0 (mountain height) results in the $\hat{H}$ *values moving up and down on the x-axis.*

The following figures show how the value of $\hat{H}$ *varies when the upper layer of ΔN varies between the layers of approximately height of 855 m asl, 1000 m asl, and 1172 m asl.*

[Figure]

The following figures show how the value of $\hat{H}$ *varies when the used to calculate the tangential wind speed is varied between layers of approximately height of 502 m asl, 582 m asl, and 677 m asl.*

[Figure]

**Detailed comments**

l.88       **as follows?**

Fixed

**Fig. 1       wouldn't it make sense to indicate which of the two is 'A' and which is 'B'?**

Figure 1 is updated with an inserted map in the lower left corner indicating which turbines are A and which are B.

l.141 **I think the Brunt-Väisälä frequency may not be well known to the audience of this journal and should therefore be defined (including its meaning).**

We have included the Brunt-Väisälä frequency in our revised manuscript (Eq. 3).

l.144 **the equal sign should be replaced by 'approximately equal'**

Fixed

l.**155       'It is assumed that the airflow that interacts directly with the mountain is spanning from the surface and up to a height of about 1 km': based on what is this assumption being made? Can the authors elaborate?**

We have assumed that the mountain will have an impact on the stable layer closest to the mountain top and have assumed that this layer has a depth of about 1 km based on the typical stable layer depth in the selected cases. We have updated the revised manuscript with the following text in line 176:

*"It is assumed that the mountain will have an impact on the stable layer closest to the mountain top and this layer is taken to have a depth of about 1 km, based on the typical stable layer in the selected cases."*

l.156       **In the EAR5 specifications (https://confluence.ecmwf.int/display/UDOC/L137+model+level+definitions) the first model level (which is labelled 137) is at 10 m – and the second (which the authors probably mean) at 31.0 m. The level closest to 1000 m (l. 157) would then be #118 (which is 'number 19' from the surface).**

Thank you for bringing this to our attention. We have corrected the information regarding what model level we have used. These are model level 120 (17) and model level 127 (10). The altitudes for each model level described in the manuscript (m asl) are correct and correspond to the table you refer to when the terrain elevation in location P (about 300 m) is subtracted.

l.158       **according to the same specification from above, the model level closest to about 550 m, would be #123, which is the 14th level from the surface**

Please see the answer above.

**Fig.3, caption 'from sea level (green) to 1500 m asl (white)': the figure (and the color bar' suggest that the color convention is the other way around....**

We have revised the caption to make it clearer for the reader that both the surrounding ocean and the highest altitudes (1500 m asl) are white.

*"The WRF domain configuration, D01, D02 and D03, with terrain elevations within each domain, ranging from sea level (green) to 1500 m asl (white). The ocean surrounding the land area is also white."*

**l.182 ,boundary conditions': what is the type of boundary conditions? The vertical (if the top level is at 50 hPa) is of particular interest. Also, is there any Rayleigh damping layer invoked, as it is usually found necessary to absorb reflection of gravity waves (e.g., Klemp et al. 2008, https://doi.org/10.1175/2008MWR2596.1)? A damping layer is quite standard the numerical investigation of mountain flows – and if the topic is mountain waves it seems particularly apropriate.**

This is correct, the upper boundary is at 50 hPa which is also stated in line 194 in the manuscript. The model is configured without Rayleigh damping, and we do acknowledge that this may be unfortunate. However, we do not expect a substantial impact on the near-surface simulation results (based on WRF best practice: WRF-dynamics-Klemp.pdf regarding Gravity-wave absorbing layer). Furthermore, we have not experienced any numerical instabilities with this model configuration. We have added this information to line 194.

**l.228 ...is the dominant mechanism....**

Fixed.

**l.233 ,....remaining consistent': usually we add '(not shown') when citing such a finding that is not demonstrated.**

Fixed

**l.272 considerably higher.**

Fixed

**Fig 6 according to what was stated before, I assume that this is the distribution for only the SE events, right? My 'doubt' seems to show that it might be good to explicitly state this again.**

This is correct. To clarify this for the readers we have include the following sentence in the introduction of Sect. 3.2 in our revised manuscript line 291:

*"These results include only wind and power production for the SE-events selected as described in Sect. 2.1"*

**Tab 2, caption: the statistical measures have to be attributed to WRF simulations. Also Bias and MAE have units (which must be given in the title row). This would also make it clear whether they refer to the capacity factor or to wind speeds.**

Thank you for this observation. We have added the units to the table.

**l.283 ….'produce 51% and 21% more…': I am not familiar with the capacity factor, but these percentages seem to be based on the respective lower value (a well-known way to make your increase to look bigger). Assuming that the capacity factor is somehow based on a maximum achievable amount, I think a more appropriate way to characterize the production increase would be 25% and 13%, respectively.**

We have added the following to the revised manuscript (line 312)

*"The capacity factor is a commonly used parameter to evaluate how well-sited a wind turbine is, and is defined as the ratio of the actually energy produced and the maximum possible energy output over the same period."*

The percentage is the capacity factor of A3 divided by the capacity factor of A2 (or A1). Which really is just the energy output from A3 divided by the energy output of A2 (A1).

**l.404 '….while underestimating….': wouldn't this suggest that WRF is not perfectly reproducing the waves (or the effect of the waves)?**

This is correct and discussed in discussed in the paragraph comparing the wind speed distributions in Fig. 7 (revised manuscript). We have also added the following to line 357 in the revised manuscript:

*"The larger negative bias at A3 suggests that the model is not fully able to reproduce the accelerated downslope winds."*

**l.320 at the end of this 'model evaluation paragraph', I am a little disappointed to see only 'mean biases', etc. If the claim is that the velocity differences are due to the formation of downslope windstorm conditions, wouldn't it be interesting to investigate whether the critical level has formed (we can get that from the model….) – maybe even with a distinction of different cases (e.g., wind direction sectors, see above, but also strong vs not so strong underestimation)?**

We do absolutely agree that this is interesting. Based on the results we have added in Sect. 3.1 at the reviewers request, we encourage further investigations into different SE-sectors in a subsequent study, as well as the presence of critical levels.

**Fig. 9 caption: '....the dotted contours....' should read 'the gray solid contours. Also: '....indicated by the dotted line in the figure c)' should read 'indicated by the full line in panel c)'.**

Fixed

**l.405 again the dashed grey contour lines...**

Fixed

**l.407 the units for wind speed are m per seconds, not meters per seconds squared**

Fixed

**l.413 not a dotted line....**

Fixed

**l.415/417 again wrong units for wind speed**

Fixed

**Fig.10, caption : please indicate which hours are displayed in panel a)**

Fixed.

**l.434 in a similar manner as ....**

Fixed

**l.454 winds speed units.....also l.459, l.460**

Fixed

**l.470 the amplitude of what is growing? And the sentence does not seem to be complete... what is 'and upstream tilting' referring to? The amplitude is tilting upstream?**

Thank you for making us aware of this.

In the revised manuscript this is changed to *"...the amplitude of the wave is growing and the wave is propagating with a tilt upstream up to about 1500 m asl"*

---

## Author Response (AR2)

**Response to Reviewer #1 and #2**

We would like to thank both the reviewers for their thorough and constructive evaluation of our manuscript. We have addressed all comments and made the necessary changes in the revised manuscript. Point-by-point responses to each comment from the reviewers are included in this document.

Sincerely,

Kine Solbakken, Eirik Mikal Samuelsen and Yngve Birkelund

Author response Report #1

- **Please add a blank space before the reference to Baines and Smith 1993 on line 98.**

  This has been corrected in the revised manuscript.

- **Same for the reference to Reinecke and Durran 1998 on line 175.**

  This has been corrected in the revised manuscript.

- **Line 183, concerning model levels 120 and 127. Here it might be beneficial to recall that ECMWF model levels are numbered top-down.**

  Thank you for this comment. The following has been added to the revised manuscript (line 178 in manuscript with tracked changes): *"..., noting that the ECMWF model levels are numbered from the model top downward."* In addition, the abbreviation ECMWF has also been included in line 144.

- **Line 305: The symbols U and N should probably be in math font.**

  This has been corrected in the revised manuscript.

- **l.237 '...propagate vertically, until they break....'**

  This has been corrected in the revised manuscript.

- **l.284 '...U in Eq. 2 is approximated...'**

  This has been corrected in the revised manuscript.

- **l. 287 '...choice of the model level.'**

  This has been corrected in the revised manuscript.

- **l.304 '....non-dimensional mountain height...'**

  This has been corrected in the revised manuscript.

- **l.313 '...whether similar relationships exist....', or '...whether a similar relationship exists...'**

  This has been corrected to *"..whether a similar relationship exists.."*

- **l.334 '...curve exists ...'**

  This has been corrected in the revised manuscript *"..the power curve consist of..."*

- **l.353 '.... to their maximum...'**

  This has been corrected in the revised manuscript.

- **Table 2, caption One assumes that MAE, Bias and R refer to observed and modelled wind speeds, and this should probably be mentioned in the caption**

  Thank you for pointing this out. The table caption in the revised manuscript has been updated to : *"...as well as the Bias, the MAE, and the correlation coefficient R between the observed and simulated wind speeds."*

- **l.375 '…Bias indicates…'**

This has been corrected in the revised manuscript.

- **l.425 '…including qualitative differences….'**

This has been corrected in the revised manuscript.

- **l.453 '….Fitch-scheme exhibits….'**

This has been corrected in the revised manuscript.

- **l.459 '….et al (2024) suggests…'**

This has been corrected in the revised manuscript.

- **l. 474 delete 'solid line' (as well)**

This has been corrected in the revised manuscript.

- **Fig.10 Caption says: 'observed vs modelled wind speeds: later (panel c), tangential windspeed is depicted. Can I trust that in panel b) it is the full wind speed (not the tangential component) which is shown? (just to compare apples to apples).**

Thank you for this comment. Panel a), b) and c) show the horizontal wind speed, while only panel d) show the tangential component. To make this clearer to the reader, we have updated the first line in the figure caption to "*a) Observed and b) simulated horizontal wind speeds at 85 m agl at the turbine locations.*" The caption for panel c) already states: "*Horizontal  wind speed at 85 m agl.*"

- **l.568 '….there is a large blue area…': please avoid describing the figures in this manner. Basically the figure shows that on the leeside of the mountain wind speed (tangential?) drops to near-zero values over a large area.**

Thank you for this suggestion. First, as noted above, panel c) shows the horizontal wind speed. To clarify this for the reader, we have updated the sentence in line 504 to "*Figure 13c) reveals large variations in horizontal wind speeds over the mountain areas*"

The sentence the reviewer is referring to has been updated in the revised manuscript to "*On the lee side of the mountain the horizontal wind speed drops to near-zero values over a large area*."

- **l. 587 'A third explanation…'**

This has been corrected in the revised manuscript.